# Breakup of nocturnal low-level stratiform clouds during the southern West African monsoon season

Maurin ZOUZOUA[1], Fabienne LOHOU[2], Paul ASSAMOI[1], Marie LOTHON[2], Véronique YOBOUE[1], Cheikh DIONE[3], Norbert KALTHOFF[4], Bianca ADLER[4], Karmen BABIĆ[4], Xabier PEDRUZO-BAGAZGOITIA[5] and Solène DERRIEN[2]

[1]Laboratoire des Sciences de Structure de la Matière, de l'Environnement et de l'Energie Solaire (LASMES)
, Université Félix Houphouët Boigny, Abidjan, Côte d'Ivoire
[2]Laboratoire d'Aérologie, Université de Toulouse, CNRS, UPS, Toulouse, France
[3]African Centre of Meteorological Applications for Development, Niamey, Niger
[4]Institute of Meteorology and Climate Research, Karlsruhe Institute of Technology (KIT), Karlsruhe, Germany
[5]Meteorology and Air Quality Group, Wageningen University and Research, Wageningen, the Netherlands

*Correspondence to*: Maurin ZOUZOUA (maurin.zouzoua@aero.obs-mip.fr)

**Abstract.**

Within the framework of the DACCIWA (Dynamics-Aerosol-Chemistry-Cloud-Interactions over West Africa) project, and based on a field experiment conducted in June and July 2016, we analyze the daytime breakup of continental low-level stratiform clouds in southern West Africa. We use the observational data gathered during twenty-two precipitation-free occurrences at Savè, in Benin. Our analysis, which starts from the stratiform clouds formation, usually at night, focuses on the role played by the coupling between cloud and surface in the transition towards shallow convective clouds during daytime. It is based on several diagnostics, including the Richardson number and various cloud macrophysical properties. The distance between the cloud base height and lifting condensation level is used as a criterion of coupling. We also make an attempt to estimate the most predominant terms of the liquid water path budget in early morning.

When the nocturnal low-level stratiform cloud forms, it is decoupled from the surface, except in one case. In early morning, the cloud is found coupled with the surface in nine cases and remains decoupled in the thirteen other cases. The coupling, which occurs within the four hours after cloud formation, is accompanied by cloud base lowering and near-neutral thermal stability in the subcloud layer. Further, at initial stage of the transition, the stratiform cloud base is slightly cooler, wetter and more homogeneous in coupled cases. The moisture jump at the cloud top is usually found to be lower than 2 g kg$^{-1}$, and the temperature jump within 1-5 K, which is significantly smaller than typical marine stratocumulus, and explained by the monsoon flow environment in which the stratiform cloud develops over West Africa. No significant difference in liquid water path budget terms was found between coupled and decoupled cases. In agreement with previous numerical studies, we found that the stratiform cloud maintenance before sunrise results from the interplay between the predominant radiative cooling, entrainment and large scale subsidence at its top.

Three transition scenarios were observed, depending on the state of coupling at initial stage. In coupled cases, the low-level stratiform cloud remains coupled until its breakup. In five of the decoupled cases, the cloud couples with the surface as

the lifting condensation level rises. In the eight remaining cases, the stratiform cloud remains hypothetically decoupled from the surface throughout its life cycle, since the height of its base remains separated from the condensation level. In case of coupling during the transition, the stratiform cloud base lifts with the growing convective boundary layer roughly between 06:30 and 08:00 UTC. The cloud deck breakup, occurring at 11:00 UTC or later, leads to the formation of shallow

convective clouds. When the decoupling subsists, shallow cumulus clouds form below the stratiform cloud deck between 06:30 and 09:00 UTC. The breakup time in this scenario has a stronger variability, and occurs before 11:00 UTC in most cases. Thus, we argue that the coupling with the surface during daytime hours has a crucial role in the low-level stratiform cloud maintenance and its transition towards shallow convective clouds.

**Keywords:** Stratiform cloud breakup, surface coupling, liquid water path budget, DACCIWA experiment.

**1 Introduction**

Low-level stratiform clouds (LLSC) are one of Earth's most common cloud types (Wood, 2012). During the West African monsoon season, LLSC form frequently at night over a region extending from the Guinean coast to several hundred kilometres inland (van der Linden et al., 2015), which includes the coastal, Sudanian and Sudanian-Sahelian climatic zones

(Emetere, 2016). The LLSC coverage persists for many hours during the following day, reducing the incoming solar radiation, and impacting the surface energy budget and related processes, such as the diurnal cycle of the atmospheric boundary layer (ABL) (Schuster et al., 2013; Adler et al., 2017; Knippertz et al., 2017). However, the diurnal cycle of those clouds is still poorly represented in numerical weather and climate models, especially over West Africa (Hannak et al., 2017). Their lifetime is generally underestimated in numerical simulations, causing high incoming solar radiation at the

surface in this region, where meteorological conditions are governed by convection activities and surface thermal and moisture gradients (Knippertz et al., 2011). This could be an important factor for which forecasts of West African monsoon features still have a poor skill (Hannak et al., 2017). Therefore, a better understanding of the processes behind LLSC over southern West Africa (SWA) would be useful for improving the quality of numerical weather prediction and climate projection. Due to a limited weather monitoring network over West Africa, the first studies addressing LLSC over this

region were mostly conducted with satellite images and traditional synoptic observations (Schrage and Fink, 2012; van der Linden et al., 2015), as well as with numerical simulations at regional scale (Schuster et al., 2013; Adler et al., 2017; Deetz et al., 2018). They emphasized that the physical processes, spanning from local to synoptic scales, such as horizontal advection of cold air associated with the West Africa monsoon, lifting induced by topography, gravity waves or shear-driven turbulence, are relevant for LLSC formation at night. However, LLSC evolution after sunrise has received

little attention in previous literature, further motivating the present study.

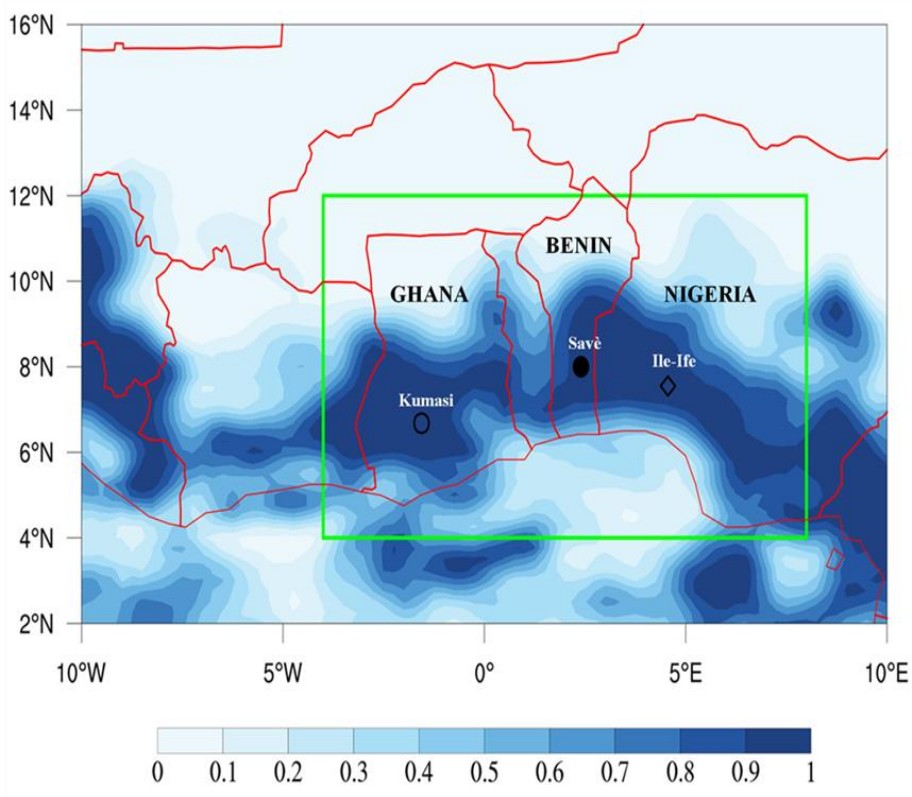

Figure 1: Low-level cloud fraction over West Africa from ECMWF (European Centre for Medium range Weather Forecast) ERA5 re-analyses (Copernicus Climate Change Service, 2019), averaged between 05:00 and 07:00 UTC on 8 July 2016. The fraction varies from 0 (clear sky) to 1 (totally covered sky). The red lines represent the geopolitical boundaries. The green box delimits the area of interest during the DACCIWA field campaign. The black markers indicate the geographical locations of the DACCIWA ground supersites, Savè in Benin (filled circle), Kumasi in Ghana (unfilled circle) and Ile-Ife in Nigeria (unfilled diamond).

During the boreal summer of 2016, a field campaign was conducted over SWA within the framework of the European Dynamics-Aerosol-Chemistry-Cloud Interaction in West Africa (DACCIWA) project (Knippertz et al., 2015). The project was developed to study the impact of increasing air pollution on SWA weather and climate. **A joint measurement**
5    **campaign took place using airborne and ground-based platforms (Flamant et al., 2017; Kalthoff et al., 2018)**. The area of interest during this field experiment is indicated in Fig. 1, which gives an example of LLSC horizontal extent between 05:00 and 07:00 UTC on 8 July 2016. One of the primary goals of this project was to provide the first high-quality and comprehensive dataset for a highly detailed study of LLSC. To this end, three so-called "supersites", which gather a large set of complementary instruments, were installed at Kumasi (6.68° N, 1.56° E) in Ghana, Savè (8.00° N, 2.40° W) in Benin, and
10   Ile-Ife (7.55° N, 4.56° W) in Nigeria (Fig. 1). The comprehensive dataset acquired at the Savè supersite paved the way for the first research studies of LLSC over SWA based on high temporal resolution observations. Adler et al. (2019) and Babić

et al. (2019a,b) studied the physical processes which govern LLSC formation and maintenance up to the next day. Dione et al. (2019) performed a statistical analysis on LLSC characteristics and low-troposphere dynamic features during the DACCIWA field campaign. The findings of these studies have been generalized and synthesized by Lohou et al. (2020) who also quantified the impact of LLSC on the surface energy budget terms for the first time. These observation-based studies focused mainly on mechanisms involved in LLSC formation during the West African monsoon season, in order to evaluate the hypotheses proposed by earlier research. **They confirmed that the horizontal advection of colder air from the Guinean coast and mechanical turbulent mixing below the nocturnal low-level jet (NLLJ) are among the main drivers for LLSC formation.** The NLLJ is one of the main features of the West African monsoon season (Parker et al., 2005; Lothon et al., 2008). The LLSC deck breakup after sunrise, which leads to a transition towards shallow convective clouds, has not yet been well documented with the unique DACCIWA dataset. Only Pedruzo-Bagazgoitia et al. (2020) have analyzed this transition by using idealized Large Eddy Simulations (LES), inspired by data collected during the LLSC occurrence on 25-26 June 2016 at Savè supersite. This was the first LES of the stratocumulus to shallow cumulus (Sc-Cu) transition over land in SWA.

Our study analyzes the transition from LLSC to shallow convective clouds of twenty-two cases observed at the Savè supersite during the DACCIWA experiment. **The results should provide complementary guidance for a numerical model evaluation of Sc-Cu transition over SWA.** The rest of this paper is organized as follows. Section 2 presents a brief state of our knowledge on the diurnal cycle of LLSC, covering SWA, and stratocumulus at other places around the world with a focus on the Sc-Cu transition. Section 3 describes the observational data and deduced diagnostics used to monitor LLSC evolution. It also presents an overview of how the contributions of some processes involved in the LLSC diurnal cycle are derived from measurements. Section 4 presents characteristics of the LLSC just before sunrise at initial stage of the transition. The relative contributions of physical processes governing the LLSC dynamic are estimated. In section 5, the LLSC evolution during daylight hours is analyzed. Finally, a summary and conclusion are given in section 6.

## 2 Review

The diurnal cycle of LLSC over SWA consists of four main stages: the stable, jet, stratus and convective phases (Babić et al., 2019a; Lohou et al., 2020). The increase of relative humidity (Rh) leading to saturation and LLSC formation is due to a cooling within the monsoon layer, up to around 1.5 km above ground level (a.g.l.), which mainly occurs during the stable and jet phases. The main process behind this cooling is the horizontal advection of cooler air from the Guinea coast, due to the combination of a maritime inflow (MI) (Adler et al., 2017; Deetz et al., 2018) and the NLLJ (Schrage and Fink, 2012; Dione et al., 2019). The onset time and strength of NLLJ, as well as the level of background humidity in the monsoon layer, are crucial for LLSC formation (Babić et al., 2019b). Indeed, from two case studies, Babić et al. (2019b) showed that weaker and later NLLJ onset leads to reduced cooling, such that saturation within the ABL may not be reached. The LLSC formation marks the end of the jet phase and the beginning of the *stratus phase*. At first, the LLSC base is located around the

NLLJ core, where cooling is at its maximum (Adler et al., 2019; Babić et al., 2019a; Dione et al., 2019; Lohou et al., 2020). During the *stratus phase*, the maximum wind speed in the NLLJ core is reduced and shifted upward by the turbulent mixing induced by longwave radiative cooling at the LLSC top, typically characteristic of stratocumulus clouds. In addition, dynamical turbulence underneath the NLLJ and convective turbulence due to the cloud-top radiative cooling are potential

drivers of coupling between the LLSC layer and the surface (Adler et al., 2019; Lohou et al., 2020). This dynamical turbulence could also be an important factor for additional cooling below the LLSC base (Babić et al., 2019a). When the LLSC deck is coupled to the surface, its base coincides quite well with the surface-based lifting condensation level (LCL) (Adler et al., 2019; Lohou et al., 2020). The final *convective phase* of the LLSC diurnal cycle starts after sunrise, when the surface sensible heat flux becomes larger than 10 W m$^{-2}$, and ends upon the LLSC breakup (Dione et al., 2019; Lohou et al.,

10   2020).

**A comprehensive overview of the current state of research on the properties and dynamic of stratocumulus clouds is presented by Garratt (1994) and Wood (2012).** Stratocumulus clouds are regulated through feedbacks between several processes: radiation, precipitation, turbulence fluxes of moisture and heat at the cloud base, entrainment and large-scale subsidence at the cloud top. The cloud Liquid Water Path (LWP) budget is considered to disentangle the respective

contribution of each process. At night, longwave radiative cooling at the stratocumulus top is the leading process governing its maintenance. **This cooling occurs because the cloud droplets emit more infrared radiation towards the free troposphere than they absorb downwelling longwave radiation from the overlying atmosphere.** The longwave cooling at the stratocumulus top is modulated by cloud-top temperature, cloud optical thickness, and thermodynamic as well as cloudy conditions in the free-troposphere (Siems et al., 1993; Wood, 2012; Christensen et al., 2013; Zheng et al., 2019).

After sunrise, solar radiation comes into play, warming the cloud and penetrating more and more down to the earth's surface as cloud layer breaking occurs. The LES performed by Ghonima et al. (2016) revealed that the effect of turbulent fluxes at cloud base depends on the surface Bowen ratio ($B$), where $B$ is the ratio of surface sensible flux to latent flux. Low values of $B$ contribute to cloud layer humidification, favouring cloud persistence. In contrast, the predominance of surface sensible heat over latent heat flux ($B > 1$) warms the cloud, leading to its evaporation. **Precipitation formation, large-scale**

**subsidence and entrainment typically warm and dry out the stratocumulus clouds (Wood, 2012; van der Dussen et al., 2016).**

The Sc-Cu transition in other climatological regions was the subject of several studies, most of which were performed over the ocean (e.g. Bretherton et al., 1999; Duynkerke et al., 2004; Sandu and Stevens, 2011; van der Dussen et al., 2016; de Roode et al., 2016; Mohrmann et al., 2019; Sarkar et al., 2019), and a few over land (e.g. Price, 1999; Ghonima et al.,

2016). **In these studies, the stratocumulus is initially coupled to the surface, with convective turbulence produced by the cloud-top radiative cooling. Specific mechanisms leading to the stratocumulus breakup are proposed, but are still based on an enhancement of the entrainment warming and drying effect.** Over land, the main driver is the intensification of convective turbulence within the ABL by solar heating at the surface (Price, 1999; Ghonima et al., 2016).

The LES developed by Pedruzo-Bagazgoitia et al. (2020) provide insight into the evolution of a coupled LLSC to the surface in terms of involved processes in the SWA monsoon conditions. **Before sunrise, the longwave radiative cooling at the LLSC top is the sole source term of the LWP budget and the primary factor maintaining this cloud layer. The breakup of the LLSC deck five hours after sunrise is primarily due to a decrease of cloud-top radiative cooling together with an increase of cloud-top entrainment.** About thirty minutes before the breakup time, a negative buoyancy flux at the LLSC base decouples it from the surface. Later, shallow cumulus clouds fully coupled to the surface appear at the convective ABL top. Since the LES performed by Pedruzo-Bagazgoitia et al. (2020) are initialized and evaluated with atmospheric and surface conditions measured at the Savè supersite, some simplifying assumptions used in our study are based on their results, and the simulated and observational results are compared.

## 3 Data and Methodology

The period in which the DACCIWA field experiment took place (June-July 2016) was divided into four synoptic phases by Knippertz et al. (2017), based on the north-south precipitation difference between the coastal and Sudanian-Sahelian areas. The first phase, the pre-onset phase, ends on 16 June 2016 with a northward shift of rainfall maximum, indicating the settlement of the West African monsoon season (Fitzpatrick et al., 2015). The second synoptic phase, the post-onset phase, characterized by higher rainfall over the Sudanian-Sahelian area, lasted from 22 June to 20 July 2016. During the first days of this phase, namely from 27 June to 8 July 2016, undisturbed monsoon flow and an increase of low-level cloudiness were observed over SWA, especially over the DACCIWA investigated area. **Between 9 and 16 July 2016, the formation of nocturnal LLSC over SWA was inhibited by drier conditions in the monsoon layer due to an unusual anticyclonic vortex (identified at 850 hPa)**. This vortex had its centre in the Southern Hemisphere (Knippertz et al., 2017; Babić et al., 2019b). During the third phase, from 21 to 26 July 2016, the rainfall maximum shifts back to the coastal area and strong westerly flow was observed in the low-troposphere over Sudanian-Sahelian area. Finally, during the final synoptic phase called the recovery phase, meteorological conditions return to a more typical behaviour for the monsoon season, with a precipitation maximum in the Sahelian region and a low-troposphere dynamic similar to the beginning of post-onset phase.

The DACCIWA supersites were located at roughly the same distance from the Guinean coast (200 km in land, Fig. 1), between the coastal and Sudanian areas, but with a different topography (Kalthoff et al., 2018). The supersites are part of the savannah ecosystem, where grassland is intercut with crops and degraded forest. Using ground-based data, Kalthoff et al. (2018) provide an overview of the low-troposphere diurnal cycle at these three ground sites. The DACCIWA field campaign includes fifteen intensive observation periods (IOPs) during which the temporal resolution of radiosondes performed at the supersites, especially at Savè, was improved. Each IOP lasted from 17:00 UTC on a given day (day-D) to 11:00 UTC on the following day (day-D+1).

The ground-based data acquired at the Savè supersite, upon which our investigation is based, offer nearly continuous information on atmospheric conditions. We analyzed a set of twenty-two LLSC occurrences for which the cloud forms at

night and persists at least until sunrise the next day. These cases have been selected over the period from 19 June to 31 July 2016 because of good data coverage (Dione et al., 2019). Only cases for which the stratus phase, determined by the methodology of Adler et al. (2019), started before 04:00 UTC on day-D+1 have been selected. Additionally, for each selected cases, no or only light precipitation (i.e. less than 1 mm) was recorded at the surface from 21:00 UTC on day-D to
16:00 UTC on day-D+1. Among these twenty-two cases, nine are IOPs, including the 07-08 July 2016 (IOP8) case (Babić et al., 2019a) and the 25-26 June 2016 case (IOP3) (Pedruzo-Bagazgoitia et al., 2020). About 60% of the selected cases occurred between 26 June and 11 July 2016, a period which falls roughly within the three first weeks of post-onset phase, and is characterized by a low-troposphere dynamic typical for the West African monsoon season. Note that we hereafter consider UTC time rather than Benin local time (UTC + 1 hour).

## 3.1 Instrumentation

Two complementary and co-located instruments installed at the Savè supersite were used to provide information on the macrophysical characteristics of LLSC (Handwerker et al., 2016): a ceilometer for the cloud base height (CBH), and a cloud radar for the cloud top height (CTH).

Through backscatter vertical profiles measured by the ceilometer, from surface to 15 km a.g.l with 15 m vertical
resolution, manufacturer software automatically provides three estimates of CBH each minute, allowing the detection of several cloudy layers. As our focus is on LLSC (the lowest cloudy layer), we use only the lowest value (hereafter CBHs). The LLSC top heights (CTHs) are derived from 5-min averaged radar reflectivity vertical profiles from 150 m to 15 km a.g.l at a vertical resolution of 30 m, by a methodology described in Babić et al. (2019) and Adler et al. (2019). According to Dione et al. (2019), the LLSC top evolves overall under 1200 m a.g.l. To be consistent with this outcome, an upper limit of
1200 m a.g.l was applied to CTHs. Unfortunately, several values of CTHs are missing, particularly during the daytime for many selected cases, due to the retrieval technique limitation.

The thermodynamical and dynamical characteristics of the low-troposphere are retrieved from radiosondes of the MODEM radiosounding system. The MODEM radiosonde collects every second (which corresponds to a vertical resolution of 4-5 m) the air temperature and relative humidity, as well as the probe GPS localization, from which horizontal wind speed
components, altitude and air pressure are deduced (Derrien et al., 2016). The sensors' accuracy is 0.2 °C, 2 % and 0.01 m for temperature, relative humidity and GPS localization, respectively. A standard radiosonde was launched every day at 05:00 UTC and usually rose to 14 km a.g.l. On IOP days, three additional radiosondes were performed at 23:00 UTC on day-D, and at 11:00 and 17:00 UTC on day-D+1. **In between these soundings, so-called "reusable" radiosondes were launched more frequently, at regular time intervals. At the height of 1.5 km a.g.l, the reusable radiosonde is released from its**
**ascending balloon, falls at the surface within a reasonable distance to be easily found and used again (Legain et al., 2013). This system allowed providing a higher temporal resolution of the conditions within the monsoon layer.** During the first six IOPs of DACCIWA, the frequent soundings were performed hourly and each 1.5 h during the other IOPs. In this

study, the radiosondes data were averaged at a final vertical resolution of 50 m. Additionally, measurements of an ultra-high frequency (UHF) wind profiler are used to derive the NLLJ core height at a 15 min time interval (Dione et al., 2019).

The meteorological conditions at surface (temperature, relative humidity and pressure of the air at 2 m a.g.l), and some terms of the surface energy budget (net radiative flux ($R_{n0}$) sensible heat ($SHF_0$) and latent heat ($LHF_0$) fluxes at 4 m a.g.l) were continuously acquired (Kohler et al., 2016). $SHF_0$ and $LHF_0$ are deduced from high-frequency (20 Hz) measurements processed with Eddy-covariance methods by using the TK3.11 software (Mauder et al., 2013).

### 3.2 Derived diagnostics to monitor the LLSC

We define some diagnostics to monitor evolution of the LLSC layer: the *fraction of low cloud coverage*, the *LLSC base height* and *cloud layer homogeneity*, the *link between LLSC deck and surface*, as well as two *characteristic times of LLSC evolution*. The LLSC depth would also be a key diagnostic, but its monitoring is limited by the low availability of CTHs cloud radar-based estimates during daytime. In addition, the humidity and temperature sensors aboard the radiosonde were affected by water deposition during crossing of the LLSC layer, so neither of is fully reliable for CTH estimates (Adler et al., 2019; Babić et al., 2019a).

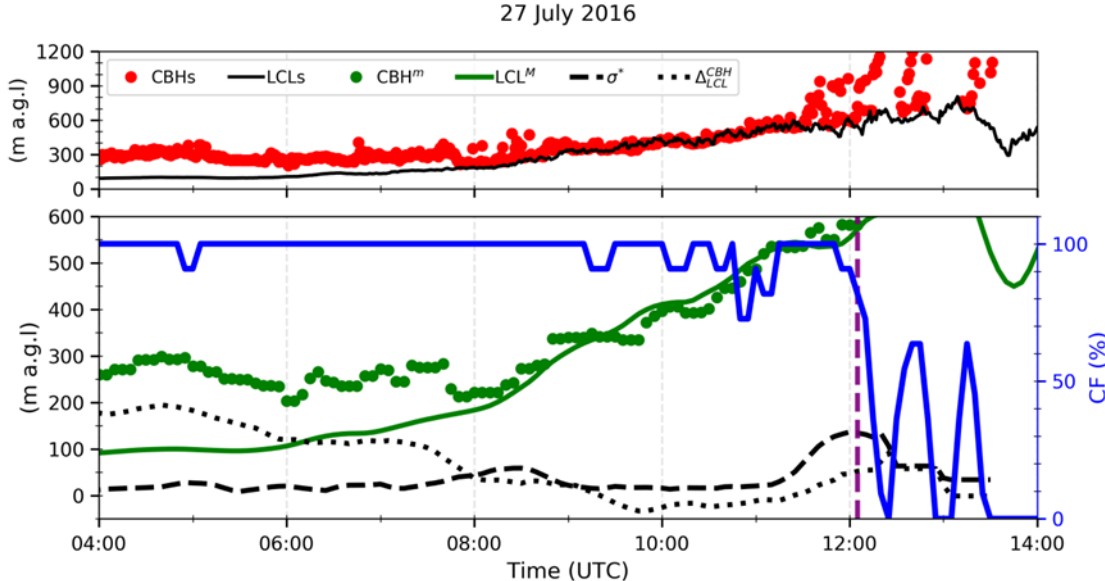

Figure 2: Time series of, 1-min ceilometer-derived CBHs and surface-based lifting condensation level (LCLs) (upper panel), and derived 5-min diagnostics (lower panel), minimum of CBHs ($CBH^m$), mean LCLs ($LCL^M$, full green line), standard deviation of the difference between CBHs and $CBH^m$ ($\sigma^*$, dashed black line), the difference between $CBH^m$ et $LCL^M$ ($\Delta_{LCL}^{CBH}$, dotted black line) and cloud coverage fraction (CF, full blue line), between 04:00 and 14:00 UTC on 27 July 2016. The vertical dashed purple line marks the breakup time of the LLSC layer ($T_b$). The Local time at Savè (in Benin) is UTC +1 hour.

**The diagnostics are calculated over a time interval of 10 minutes with a moving window of 5 minutes, which is suitable for resolving the processes-related to convection.** Figure 2 illustrates our methodology, with an example of measurements and derived diagnostics for the case of 26-27 July 2016.

- _Fraction of low cloud coverage_: The low-cloud fraction (*CF*) is defined as the percentage of 1-min ceilometer CBHs lower
than or equal to 1000 m a.g.l. Thus, a *CF* greater or equal to 90 % corresponds to the presence of LLSC. A similar methodology was used by Adler et al. (2019), but with a threshold of 600 m a.g.l. We extend the upper limit to 1000 m a.g.l to take into account the LLSC base rising during the *convective phase* (Lohou et al., 2020). On 27 July 2016 (Fig. 2), the few periods between 04:00 UTC and 11:30 UTC with CF < 90 % indicate intermittent break within the LLSC deck. This feature is common to many other cases.

- _The LLSC base height and cloud layer homogeneity:_ As seen in Fig. 2, the cloud "base height" may be more or less homogeneous in time and space, from a compact level cloud deck (like from 06:00 UTC to 06:30 UTC in Fig. 2) to a fragmented cloud layer or even separated cumulus clouds (like from 12:30 UTC to 13:00 UTC in Fig. 2). In the latter case, the ceilometer beam often hits the cumulus cloud base or higher edges, introducing a large variability of the so-called and measured "CBH" (which is here more rigorously the first height above ground with detected clouds). In order to take this
aspect into account in the LLSC base definition, and to quantify the LLSC base homogeneity, we define two other diagnostics based on 1-min ceilometer-derived CBHs. The first is a characteristic LLSC base height, defined as the minimum of CBHs over the 10-min intervals ($CBH^m$). The second is the standard deviation of CBHs (<=1000 m a.g.l) minus $CBH^m$ within the 10-min intervals ($\sigma^*$), which provides insight into the LLSC layer heterogeneity by deleting the effect of the CBH morning increase (Lohou et al., 2020). Small values of $\sigma^*$ indicate nearly constant CBHs; that is, a horizontally
homogenous cloud layer base (as from 04:00 UTC to 07:00 UTC on 27 July). High values of $\sigma^*$ indicate irregular bases of the LLSC layer or a mix of cloud base and edges after the LLSC breakup (as around 12:00 UTC on 27 July). The increase of $\sigma^*$ from 21 to 135 m after 11:00 UTC on 27 July (Fig. 2) typically indicates an evolution towards a more heterogeneous LLSC layer.

- _The link between LLSC deck and surface:_ When a LLSC layer is coupled to the surface, its base coincides rather well with
the LCL (Zhu et al., 2001; Wood, 2012). The coupling between the LLSC deck and surface may then be assessed by the distance between the cloud base height and LCL. We define $LCL^M$ as the mean value of LCL calculated on a 10-min time interval by using the formulation of Romps (2017) with near surface meteorological measurements. The coupling is estimated by $\Delta_{LCL}^{CBH} = CBH^m - LCL^M$. On 27 July 2016 (Fig. 2), $\Delta_{LCL}^{CBH}$ is initially around 190 m, from 04:00 to 06:00 UTC, indicating that the LLSC is decoupled from the surface. The progressive increase of LCL starting around 06:00 UTC leads to
LLSC coupling with the surface slightly before 08:00 UTC.

Finally, the diagnostics $LCL^M$, $\Delta_{LCL}^{CBH}$ and $\sigma^*$ defined before are smoothed with a moving average over 30 minutes every 5 min (Fig. 2).

- *Characteristic times of LLSC evolution:* From the above diagnostics, two specific times characterizing the LLSC lifetime are determined;

    • The surface-convection influence time ($T_i$) corresponding to the time from which the low-level cloud coverage reacts to solar heating at the surface. The method to determine $T_i$ depends on the evolution of LLSC during the *convective phase*. Thus, it will be precisely defined later in the text, after the presentation of the different observed scenarios.

    • The LLSC breakup time ($T_b$) which corresponds to the end of LLSC occurrence. It is the time (after 06:30 UTC) from which *CF* is lower than 90 % during at least one hour. Figure 2 (lower panel) shows several periods, between 09:00 UTC and 11:00 UTC, with *CF* lower than 90 %, but for less than one hour, so that they are included in the LLSC lifetime. For this case, $T_b$ is at 12:05 UTC.

## 3.3 LWP budget

The LWP tendency equation is based on the assumption of horizontally-homogeneous LLSC vertically well-mixed by convective turbulent mixing driven by the cloud-top radiative cooling. Following van der Dussen et al. (2014), this equation can be split into five relevant processes:

$$\frac{\partial \text{LWP}}{\partial t} = \text{BASE} + \text{ENT} + \text{PREC} + \text{RAD} + \text{SUBS} \tag{1}$$

in which

$$\text{BASE} = \rho\eta(\overline{w'q_t'}^b - \Pi\gamma\overline{w'\theta_l'}^b) \tag{1.a}$$

$$\text{ENT} = \rho w_e(\eta\Delta q_t - \Pi\gamma\eta\Delta\theta_l - h\Gamma_{ql}) \tag{1.b}$$

$$\text{PREC} = \rho\Delta\wp \tag{1.c}$$

$$\text{RAD} = \rho\eta\gamma\Delta F_{rad} \tag{1.d}$$

$$\text{SUBS} = -\rho h\Gamma_{ql}w_{s,CTH} \tag{1.e}$$

representing the effects of turbulent moisture and heat fluxes at cloud base (BASE), evaporation or condensation caused by the entrainment of ambient air from aloft (ENT), precipitation formation (PREC), radiative budget along the cloud layer (RAD) and large-scale subsidence (SUBS) at its cloud top.

In the above equations (1.a) to (1.e), $\overline{w'q_t'}^b$ and $\overline{w'\theta_l'}^b$ are respectively the total moisture specific humidity ($q_t$) and liquid-water potential temperature ($\theta_l$) heat fluxes at cloud base (superscript "b"), $\rho$ is the mean air density over cloud layer and h is the cloud depth. $\Delta F_{rad}$ and $\Delta\wp$ are the differences, in net radiation and precipitation respectively, between the cloud top and

base heights (van der Dussen et al., 2014). $\Delta\theta_l$ and $\Delta q_t$ are the jumps of respectively $\theta_l$ and $q_t$ across the cloud layer. $w_e$ and $w_{s,CTH}$ are the cloud top entrainment and large-scale subsidence velocities, respectively.

The equations also introduce following parameters: the Exner function $\Pi = \left(\frac{P}{1000}\right)^{\frac{R_d}{C_p}}$; the adiabatic lapse rate of liquid water content $\Gamma_{ql} = g\eta\left(\frac{q_s}{R_d\overline{T}} - \frac{\gamma}{C_p}\right)$; $\gamma = \frac{L_v q_s}{R_v \overline{T}^2}$ and $\eta = \left(1 + \frac{L_v\gamma}{C_p}\right)^{-1}$. In those parameters, P and $\overline{T}$ are respectively the cloud layer pressure and temperature, $q_s$ is the saturation water vapour specific humidity at P and $\overline{T}$. $R_d$ and $R_v$ are respectively the dry air and water vapour gas constant. $L_v$, $C_p$ and g correspond, respectively, to vaporization latent heat of water, specific heat of dry air at constant pressure and gravitational acceleration.

For our analysis of DACCIWA cases, we consider the LWP budget in the early morning, and use the 05:00 UTC radiosounding, ceilometer and cloud radar measurements to estimate some terms of equation (1). In fact, this is the optimal time for the assumption of horizontally homogeneous and vertically well-mixed LLSC layer. **The PREC term is typically near zero because no significant rain was measured at surface for the selected cases.** The BASE term is not estimated because the turbulent fluxes at LLSC base cannot be deduced from available dataset at the Savè supersite. According to Pedruzo-Bagazgoitia et al. (2020), the BASE term is small at this time relative to the three terms RAD, ENT and SUBS. The latter are the most significant contributions in early morning that we attempt to estimate.

$$LWP = -\frac{1}{2}\rho\Gamma_{ql}h^2 \tag{2}$$

The RAD term (Eq. 1.d) is retrieved from the vertical profiles of upwelling and downwelling radiative fluxes which are computed using the Santa Barbara DISORT Atmospheric Radiative Transfer (SBDART) model (Ricchiazzi et al., 1998). This software tool, which solves the radiative transfer equation for a plane-parallel atmosphere in clear and cloudy conditions, was used in the studies of Babić et al. (2019a) and Adler et al. (2019) to estimate temperature tendency due to radiative interactions during the LLSC diurnal cycle. For our simulations, the model configuration was very similar to that used in these studies. We prescribed 65 vertical input levels with a vertical resolution of 50 m below 2 km a.g.l, 200 m between 2 and 5 km a.g.l, and, 1 km above 5 km a.g.l. The vertical profiles of air pressure, temperature and water vapour, density as well as the integrated water vapour are based on 05:00 UTC standard radiosounding data. The cloud optical thickness, which varies with its water and ice content, is required to describe a cloud layer in the SBDART model. However, the LWP provided by the microwave radiometer deployed at the Savè supersite (Wieser et al., 2016) includes all existing cloudy layers, and is not available for five of our selected cases. Therefore, the LLSC optical thickness is determined from a parameterized LWP (Eq. 2), by assuming an adiabatic cloudy layer in which the liquid water mixing ratio ($q_l$) increases linearly (van der Dussen et al., 2014; Pedruzo-Bagazgoitia et al., 2020). The downwelling longwave radiations from potential mid-level and high-level clouds may reduce radiative cooling at the LLSC top (e.g. Christensen et al., 2013). However, the cloud layers above the LLSC (base, top and water content) cannot be precisely described in the SBDART model from the available dataset. Thus, the radiative effect of higher clouds is not directly included in our estimate of downwelling radiative fluxes, but is partially taken into account through vertical profiles of temperature and relative

humidity given by the radiosonde. As the shortwave radiations are zero before sunrise, only the longwave range, 4.5-42 μm with spectral resolution of 0.1μm (Babić et al., 2019a), was selected for radiative fluxes calculations. For all cases, the vertical optical depth of ABL aerosol is fixed at 0.38, which corresponds to the average value of measurements performed with a sun photometer in June and July 2016 at Savè.

For the ENT term (Eq. 1.b), we use the parameterization of Stevens et al. (2005) to estimate $w_e$:

$$w_e = A * \frac{\Delta F_{rad}}{\Delta \theta_l} \tag{3}$$

in which $A$ is a non-dimensional quantity representing the efficiency of warming caused by the input of free tropospheric air into the LLSC layer by the buoyancy-driven eddies generated by cloud-top radiative cooling. $A$ varies with $\Delta \theta_l$, $\Delta q_t$, wind shear at cloud top, surface turbulent fluxes and cloud microphysical processes via the buoyancy flux vertical profile (Stevens et al., 2005; Stevens, 2006). Despite the spatial and temporal variability of $A$, its value is generally fixed and treated as a constant parameter in several research studies (e.g. van Zanten et al., 1999; van der Dussen et al., 2014). The value of $A$ used in the literature varies from one study to another. By considering the results of the LES developed by Pedruzo-Bagazgoitia et al. (2020) on a DACCIWA case, just before sunrise, with $w_e \approx 4.5 \; mm.s^{-1}$, $\Delta \theta_l \approx 4 \; K$, a cloud-top longwave radiative cooling of around 43 W m$^{-2}$, and, ρ$\approx 1.13$ kg.$m^{-2}$ as the average value from surface to 1000 m a.g.l (from 26 June 05:00 UTC sounding), we obtain $A \approx 0.5$. This means that, the contribution of tropospheric air entrainment to heat budget at the LLSC top is around two times smaller than that of cloud-top radiative cooling. For the sake of simplicity, and due to the lack of a precise estimate, we assume here same behaviour for all DACCIWA cases, and consider $A = 0.5$ in our analysis.

The jumps in temperature $\Delta \theta_l$ and in total water content $\Delta q_t$ are estimated from the soundings. We write $\theta_l = \theta - \frac{1}{\Pi}\left(\frac{L_v}{C_p}\right) q_l$, with $\theta$ as the potential temperature, whereas $q_t = q + q_l$. We define:

$$\Delta \varphi \approx \varphi^+ - \varphi^- \tag{4}$$

where $\varphi$ can be either $\theta_l$ or $q_t$. $\varphi^+$ and $\varphi^-$ are in theory the values of $\varphi$ just above and below the cloud top, respectively. Under the assumption of a well-mixed cloud layer, $\theta_l$ ($q_t$) is conserved through the cloud layer and increases (decreases) abruptly in the warmer (drier) ambient air right above (vanZanten et al., 1999). Thus, $\Delta \theta_l$ and $\Delta q_t$ can be estimated from the vertical profiles of $\theta$ and $q$ derived to the 05:00 UTC standard sounding. For $\theta_l^+$ and $q_t^+$, we consider the mean over the 100 m just above CTH. For $\theta_l^-$ and $q_t^-$, we consider the sounding level just below CBH. In brief, we use:

$$\begin{cases} q_t^- = q_{t \; \{below \; cloud \; top\}} = q_{t \; \{below \; cloud \; base\}} = q_{\{below \; cloud \; base\}} \\ \theta_l^- = \theta_{l \; \{below \; cloud \; top\}} = \theta_{l \; \{below \; cloud \; base\}} = \theta_{\{below \; cloud \; base\}} \end{cases} \tag{5}$$

**For the SUBS term (Eq. 1.e), we cannot accurately estimate the large-scale subsidence velocity at LLSC top. One possibility is to compute estimates from models or re-analyses. However, we decided to discard this approach, because the subsidence vertical profiles from regional simulations with Consortium for Small-Scale Modelling (COSMO) or from ERA-interim and ERA-5 reanalyses showed a very high temporal variability and a strong lack of coherence among the different cases.** According to cloud-radar CTHs estimates, the LLSC top is often stationary at the end of stratus phases during the DACCIWA field experiment. This feature has been observed (Adler et al., 2019; Babić et al.,

2019a; Dione et al., 2019) and also simulated by Pedruzo-Bagazgoitia et al. (2020). Based on the LLSC top stationarity at the time of our LWP budget analysis, $w_{s,CTH}$ is estimated following Lilly (1968):

$$\frac{\partial CTH}{\partial t} = w_{s,CTH} + w_e \approx 0 \tag{6}$$

## 4 LLSC during the stratus phase

In this section, we document the *stratus* phase of the LLSC diurnal cycle. The aim is to analyze the way the cloud layer is coupled to surface processes, and the possible impacts of coupling on cloud characteristics (macrophysical properties and LWP terms). During the DACCIWA field campaign, sunrise occurred at Savè between 05:33 and 05:42 UTC (Kalthoff et al., 2018). According to Lohou et al. (2020), the *convective phase* starts between 07:30 and 09:00 UTC. The last radiosonde released before the *convective phase* is performed at 06:30 UTC, thus the analysis in this section concerns the period from LLSC formation (beginning of the *stratus phase*) to 06:30 UTC on day-D+1.

### 4.1 Coupled and decoupled LLSC

We first analyze the evolution of LLSC base height (CBH) and its link with the NLLJ core height and surface-based LCL along the *stratus phase* (Fig. 3). The CBH and LCL at the beginning of the *stratus phase* (Fig. 3a and b) are given by diagnostic parameters $CBH^m$ and $LCL^M$, respectively, when the LLSC forms, and the NLLJ core height is the hourly-averaged value at that time. For the end of the *stratus phase* (Fig. 3c and d), CBH, LCL and NLLJ are averaged between 04:00 and 06:30 UTC on day-D+1.

When the LLSC forms, its base is located within the NLLJ core, where cooling driven by the horizontal advection is at its maximum (Adler et al., 2019; Dione et al., 2019; Lohou et al., 2020). Both the CBH and NLLJ core height range between 50 and 500 m a.g.l (Fig. 3a) and are a hundred meters above surface-based LCL, except for one case (Fig. 3b). This means that the LLSC is decoupled from the surface when it forms.

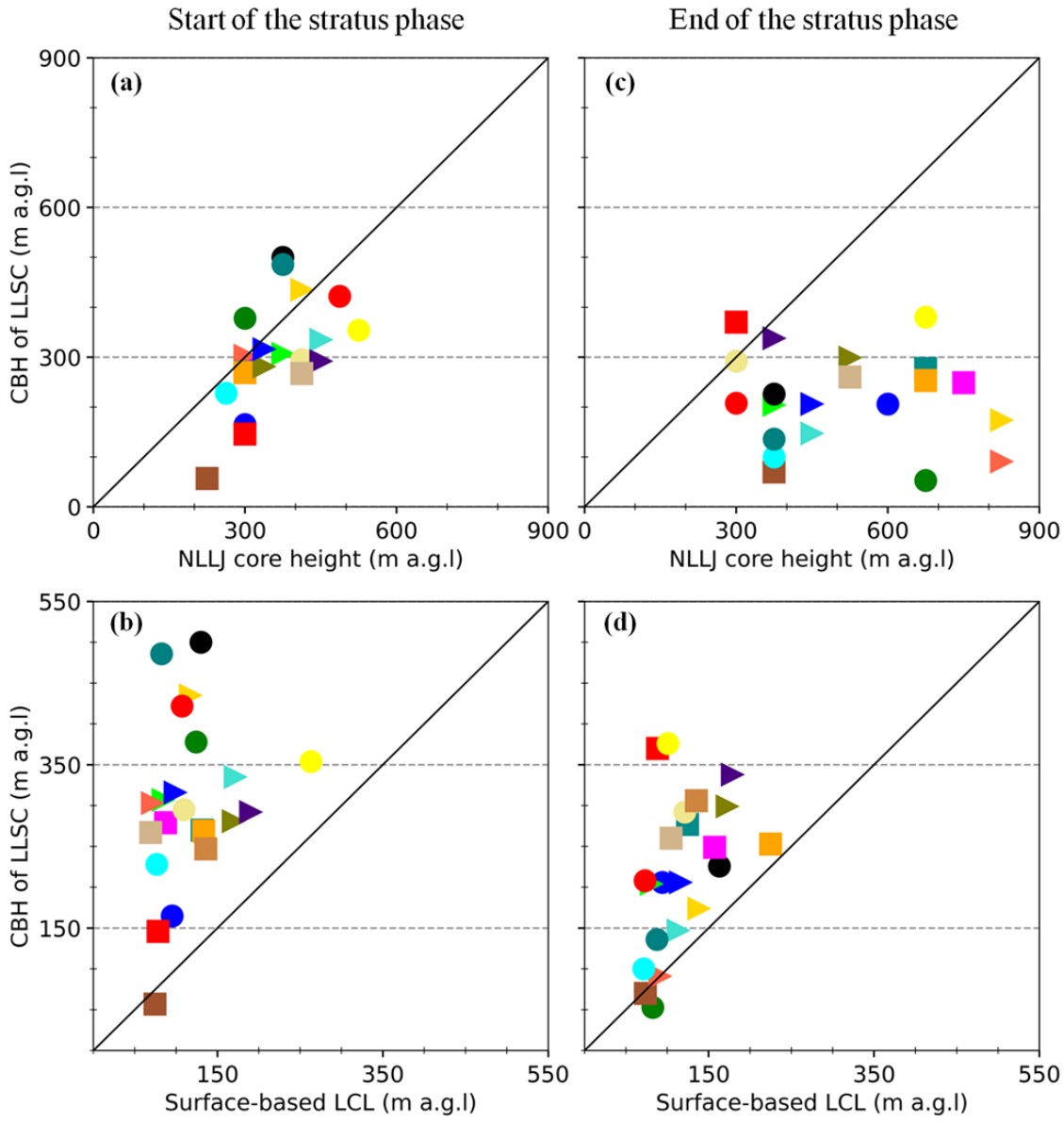

Figure 3: LLSC base height (CBH) against nocturnal low-level jet (NLLJ) core height (top panels), and surface-based lifting condensation level (LCL) (bottom panels), at the start **(a, b)** and at the end of *stratus phase* **(c, d)**. Each of the twenty-two selected cases is represented by a different marker.

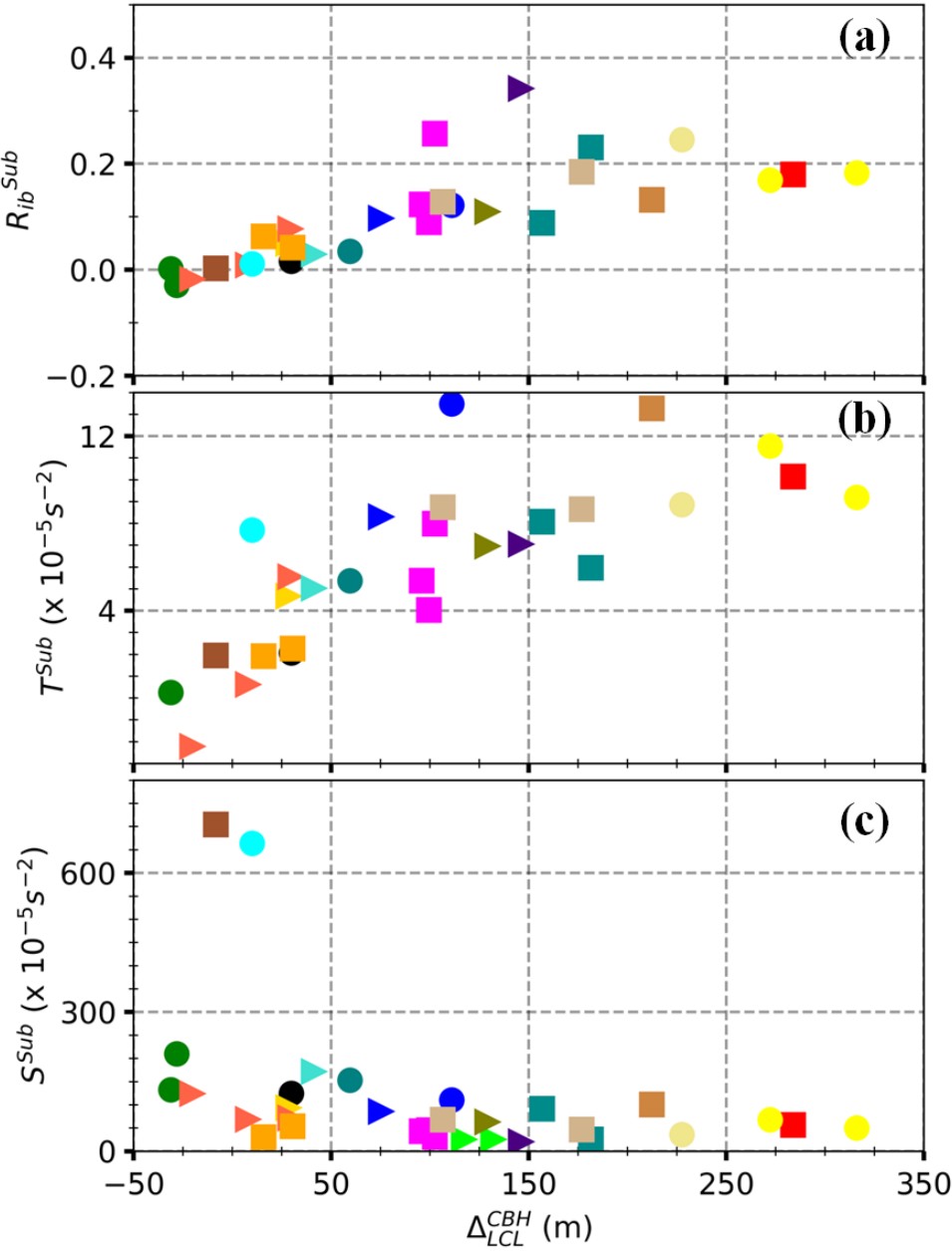

Figure 4: Bulk Richardson number ($R_{ib}^{Sub}$, **a**), and its thermal ($T^{Sub}$, **b**) and vertical wind-shear ($S^{Sub}$, **c**) composing terms, as a function of the diagnostic parameter $\Delta_{LCL}^{CBH}$, which corresponds to the mean distance between the LLSC base height (CBH) and the surface-based lifting condensation level (LCL), performed by using all radiosoundings available from 04:00 to 06:30 UTC on day-D+1 for each studied case. Each marker corresponds to one case.

At the end of the *stratus phase*, we can see that the relationship between CBH and the NLLJ core height has totally changed (Fig. 3c). There is no clear linear link between both, and CBH remains mostly lower than or equal to 300 m a.g.l, while NLLJ core height is above 600 m a.g.l in several cases. This is most likely because, during the *stratus phase*, the jet axis is shifted upward by the convective turbulence within the LLSC layer (Adler et al., 2019; Dione et al., 2019; Lohou et al., 2020). In addition to the jet axis rising, the averaged CBH decreases by the end of the *stratus phase* (Fig. 3a and c) for most cases. In some cases, CBH coincides pretty well with LCL (Fig. 3d), which indicates coupling between the LLSC layer and the surface. However, in others, CBH is still at least 100 m higher than LCL, meaning that the LLSC layer remains decoupled from the surface.

We further analyze the coupling between the LLSC deck and surface at the end of *stratus phase* by using the bulk Richardson number (Stull, 1988) of subcloud layer ($R_{ib}^{Sub}$). It reads:

$$R_{ib}^{Sub} = \frac{T^{Sub}}{S^{Sub}} \text{ with } T^{Sub} = \frac{g}{\theta} * \frac{\Delta\theta}{CBH} \text{ and } S^{Sub} = \left(\frac{\Delta U}{CBH}\right)^2. \tag{7}$$

$T^{Sub}$ and $S^{Sub}$ are respectively the thermal and horizontal wind shear contributions to Richardson number. $\frac{\Delta\theta}{CBH}$ and $\frac{\Delta U}{CBH}$ are the bulk vertical gradient of $\theta$ and horizontal wind speed (U), respectively within the subcloud layer (from surface to cloud base), with the assumption that U is null at surface. $R_{ib}^{Sub}$ is estimated with all radiosoundings available from 04:00 to 06:30 UTC on day-D+1, for each studied case. The subcloud layer height is estimated with the half-hourly median of $CBH^m$ at radiosonde released time (Eq. 7).

Figure 4 shows $R_{ib}^{Sub}$ (Fig. 4a), $T^{Sub}$ (Fig. 4b) and $S^{Sub}$ (Fig. 4c) as a function of the half-hourly median value of $\Delta_{LCL}^{CBH}$ at radiosonde released time. The smaller $\Delta_{LCL}^{CBH}$, the lower $R_{ib}^{Sub}$. Interestingly, when $\Delta_{LCL}^{CBH}$ is smaller than 75 m, $R_{ib}^{Sub}$ is less than or equal to 0.1 (Fig. 4a). This evidence suggests that the potential coupling between LLSC and surface during the *stratus phase* is driven by underlying turbulent mixing. A similar tendency was found by Adler et al. (2019), who analyzed the soundings performed along the *stratus phase* of eleven IOPs.

As $R_{ib}^{Sub}$, the $T^{Sub}$ term increases with $\Delta_{LCL}^{CBH}$, whereas the $S^{Sub}$ term is nearly constant. This means that, when CBH is close to LCL, the subcloud layer is well mixed, although the shear-driven turbulence is not particularly significant. Thus, the coupling between LLSC and surface at the end of the *stratus phase* seems to be mostly linked to thermal stratification in the subcloud layer, rather than to shear-driven turbulence.

Finally, based on Fig. 4 (a and b), the value of 75 m is used thereafter as a threshold for $\Delta_{LCL}^{CBH}$ to distinguish coupled and decoupled LLSC at the end of the *stratus phase*. Through this classification, our set of twenty-two studied cases includes nine LLSC coupled to the surface (case C) and thirteen LLSC decoupled from the surface (case D) (Table A-1). Among the nine selected IOPs, three (N° 5, 6 and 8) and six (N° 3, 4, 7, 9, 11 and 14) are cases C and D, respectively.

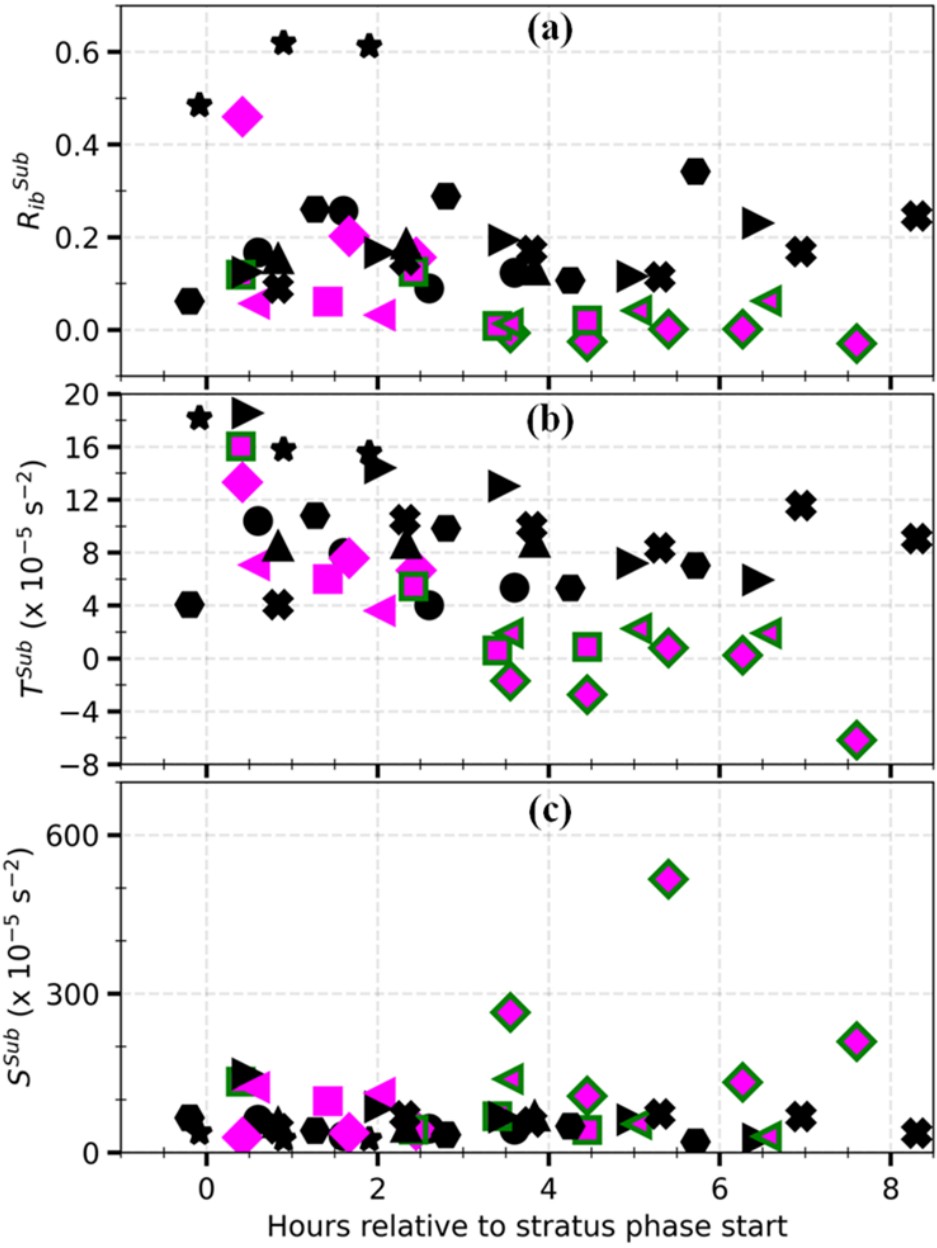

Figure 5: Evolution of the bulk Richardson number ($R_{ib}^{Sub}$, **a**) and its thermal ($T^{Sub}$, **b**) and vertical wind-shear ($S^{Sub}$, **c**) composing terms during the stratus phase, based on all the soundings available until 06:00 UTC on day-D+1 during the nine selected IOPs (Table A-1). The quantities are presented against the radiosonde released time, which is expressed in hours relative to the start of the stratus phase. Each IOP is represented by a marker. C and D stand for coupled and decoupled LLSC at the end of stratus phase respectively. The green edge for C cases indicates that the mean distance between LLSC base height and surface-based lifting condensation level (LCL) ($\Delta_{LCL}^{CBH}$) is of less than 75 m at sounding time, meaning that LLSC is coupled to the surface.

Based on reusable radiosoundings available for the nine selected IOPs, the temporal evolution of $R_{ib}^{Sub}$ and its composing terms have been calculated from the start of the *stratus phase* up to 06:30 UTC on day-D+1 (Figure 5). $R_{ib}^{Sub}$, $T^{Sub}$ and $S^{Sub}$ in C and D cases are similar when the LLSC forms. For C cases, $T^{Sub}$ decreases to zero (neutral stratification) within the three following hours, while $S^{Sub}$ remains almost constant, which causes a decrease of $R_{ib}^{Sub}$ (Fig. 5a and b). In C cases presented in Fig. 5, the definitive coupling with surface occurs within four hours after the beginning of the *stratus phase*. The same behaviour is observed for C cases, which are not IOP and therefore not included in Fig. 5 (not shown). For D cases, the subcloud layer remains thermally stable along the *stratus phase*, and shear-driven turbulence is of the same order as for C cases. Considering these results, it appears that, the shear-driven turbulence in the subcloud layer is not the main process causing the coupling of LLSC layer with the surface during the *stratus phase* in C cases.

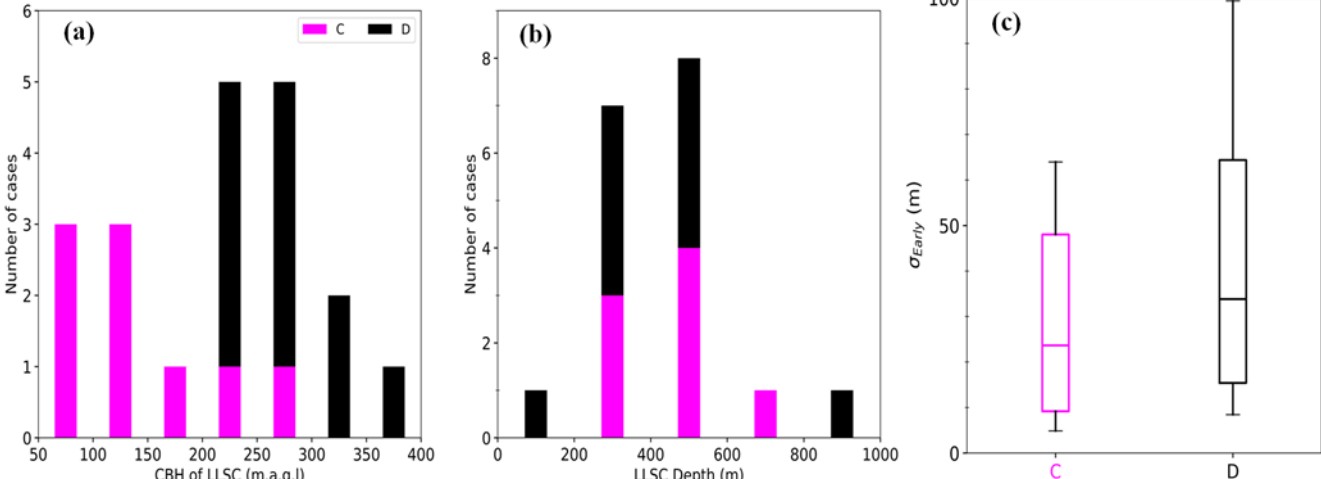

Figure 6: Statistic on the LLSC macrophysical characteristics at the end of the *stratus phase*, performed on the twenty cases (the nine cases C and eleven cases D out of thirteen), for which the LLSC is present (CF ≥ 90%) over at least 70% of the time between 04:00 and 06:30 UTC on day-D+1. Distributions of LLSC base height (CBH, **a**), the same as in Figure 3, and depth **(b)**, calculated by using the median value between 04:00 and 06:30 UTC of cloud-radar estimated CTHs as LLSC summit. The depth was not estimated for two cases (one C and one D) out of twenty due to missing CTH data. Statistical information on $\sigma_{Early}$ **(c)**, which is the median value between 04:00 and 06:30 UTC of diagnostic parameter $\sigma^{*}$, measuring the LLSC base homogeneity. The edges of the boxes represent the 25[th], median and 75[th] percentiles, and the whiskers, the minimum and maximum values. C and D stand for coupled and decoupled LLSC respectively.

In conclusion, the LLSC is typically decoupled from the surface at formation. Subsequently, its base lowers during the first hours of the *stratus phase*. In C cases, this decrease is more important and leads to coupling between the cloud deck and the surface before sunrise. The lowering of the LLSC base was first pointed out by Babić et al. (2019a) for the 07-08 July case. They explained this feature by an additional cooling in the subcloud layer, mainly due to a shear-driven turbulent mixing caused by the NLLJ. Yet, no substantial differences in wind shear below the LLSC are observed between the C and D cases, indicating that the processes related to mechanical turbulence underneath the LLSC cannot fully explain the coupling observed by the end of the *stratus phase*. The other relevant processes which may couple the LLSC to surface in night-time conditions are discussed in section 4.3. In the next paragraph, we analyze the LLSC macrophysical characteristics in C and D cases at the end of the *stratus phase*, i.e. just before the *convective phase*.

The distributions of averaged LLSC base height and depth at the end of the *stratus phase* are summarized in Fig. 6a and b, respectively. Only the twenty cases for which the cloud is persistent between 04:00 and 06:30 UTC on day-D+1 are considered (including nine C cases and eleven D cases). Note that the depth could not be estimated for two of these cases because of missing CTH data. The CBH ranges within 50-200 m a.g.l for C cases, and within 200-400 m a.g.l for D cases. This clear difference between coupled and decoupled LLSC explains the bimodal distribution of morning CBH observed by Kalthoff et al. (2018). In contrast, the morning LLSC depth does not depend on the state of coupling with the surface.

Figure 6c shows the LLSC base homogeneity at the end of the *stratus phase* by presenting statistical information about $\sigma_{Early}$, which is the median value of diagnostic parameter $\sigma^*$ between 04:00 and 06:30 UTC on day-D+1 for each considered case. The median of $\sigma_{Early}$ is 24 m for C cases and 34 m for D cases. Their 25[th] percentiles and minimums are close, but the 75[th] percentile for D cases is more than 15 meters higher than that of C cases, and the maximum is significantly larger, close to 100 m. This reveals the larger LLSC base heterogeneity found for several D cases. Likely, the coupling with surface limits fragmentation of the LLSC layer, and helps to maintain cloud base homogeneity in C cases.

In brief, the coupling mechanism favours a lower CBH and a slightly more homogeneous cloud base in coupled cases. But the LLSC depth is similar in coupled and decoupled cases, such that the LLSC vertical extension does not seem to be influenced by the coupling with the surface. **This may be related to the negligible contribution of surface fluxes during the *stratus phase* (Pedruzo-Bagazgoitia et al., 2020)**.

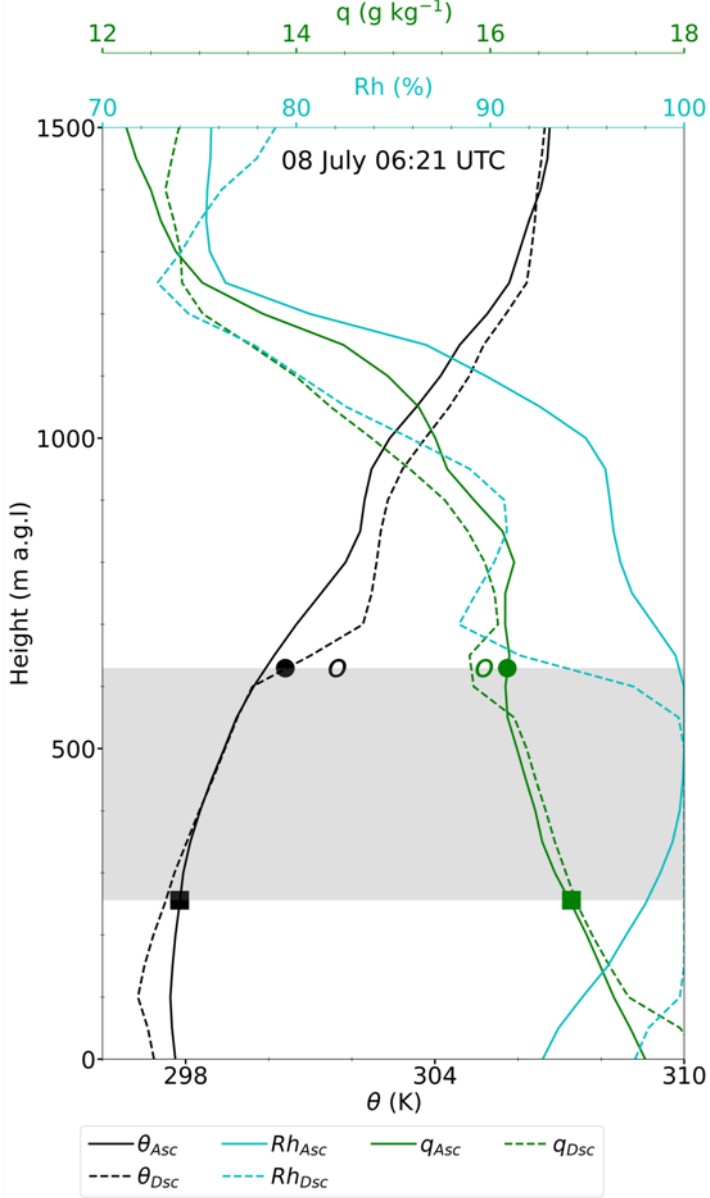

Figure 7: Vertical profiles of the low-troposphere acquired by the re-usable radiosonde of 08 July 2016 at 06:21 UTC, when the probe ascends ('Asc', filled line) and descends ('Dsc', dashed line). The variables shown are relative humidity (Rh), potential temperature (θ) and water vapour specific humidity (q). The shaded grey delimits the LLSC layer, based on ceilometer and cloud-radar measurements. The values of $\varphi^+$($\varphi^-$) (Eq. 4) for θ and q are marked with a dot (square). The filled symbols correspond to the ascent, whereas the unfilled symbols correspond to the descent.

## 4.2 LWP terms

In this sub-section, we attempt to estimate the terms of LWP budget at the end of the *stratus phase*, in order to answer several questions:

1) Using observations, do we obtain results similar to those of previous numerical simulations, particularly that of Pedruzo-Bagazgoitia et al. (2020)?

2) Does the LWP budget analysis help us to differentiate decoupled and coupled cases?

As previously seen, the most important contributions to the LWP budget are that of radiation, entrainment and subsidence. Based on available observations and by using the SBDART model, we estimate the ENT and RAD terms (Eq. 1.b and d respectively), and also give a rough order of magnitude of the SUBS term (Eq. 1.e). The LLSC layer here is defined by the averaged CBH and CTH at the end of the *stratus phase* (Fig. 6a and b).

We first discuss the jumps $\Delta q_t$ and $\Delta\theta_l$ across the cloud top (Eq. 4 and 5), which are involved in ENT term. They are estimated by using the 05:00 UTC (day-D+1) standard radiosoundings. **The liquid water build-up on the probe sensors possibly renders some measurements suspect, especially near the cloud top.** In order to evaluate the impact of this issue on our jump estimations from the 05:00 UTC standard radiosonde, we first consider a reusable sounding at a different time, for which the probe has crossed the LLSC layer at both the ascent and descent. At ascent, sensors are reliable at the cloud base, but may obtain incorrect data when they reach the cloud top. At descent, it is the reverse: accurate at the cloud top but possibly erroneous when reaching the cloud base. This is shown in Fig. 7, which displays the vertical profiles of $\theta$, q and Rh measured by the reusable sounding of 08 July 2016 at 06:21 UTC, during both the probe ascent and descent. By analyzing the Rh vertical profiles, we can see that the upper limit of the saturated layer (Rh $\leq$ 98.5 %), i.e. the top of LLSC layer, obtained by the descent measurements is more consistent with cloud radar-estimated CTH than that obtained during the ascent. Further, the descent measurements indicate warmer and drier atmospheric conditions from CTH to around 800 meters above, with $\theta^+$ ($q^+$) around 1 K (0.3 g kg$^{-1}$) higher (smaller). By analyzing all reusable soundings of that kind during daytime, we find that the maximum underestimation (overestimation) of $\theta^+$ ($q^+$) during the ascent due to wetting of the sensors is about 1.2 K (0.3 g kg$^{-1}$). The overestimation of $q^+$ by ascending sounding is within the measurement accuracy while, compared to the 0.2° C measurement accuracy, the underestimation of $\theta^+$ is significant. Consequently, we only consider a systematic error of 1.2 K on the estimates of $\theta^+$ from the 05:00 UTC standard radiosounding, for which we can only rely on the ascent (the descent is too far away from the supersite).

Figure 8 displays $\Delta q_t$ and $\Delta\theta_l$ against $q^-$ and $\theta^-$ respectively, as estimated for fourteen cases (eight C cases and six D cases) among the twenty cases in Figure 6, for which there is evidence that the radiosonde flew throughout the LLSC layer. It first reveals that the thermodynamical conditions of the subcloud layer are quite steady during this summer period, with only a 1.5 g kg$^{-1}$ and 2 K variation range for humidity and temperature, respectively, over all cases. A similar conclusion was drawn by Adler et al. (2019). This may be due to the fact that the considered cases occurred in nearly similar synoptic conditions over SWA (Table A-1).

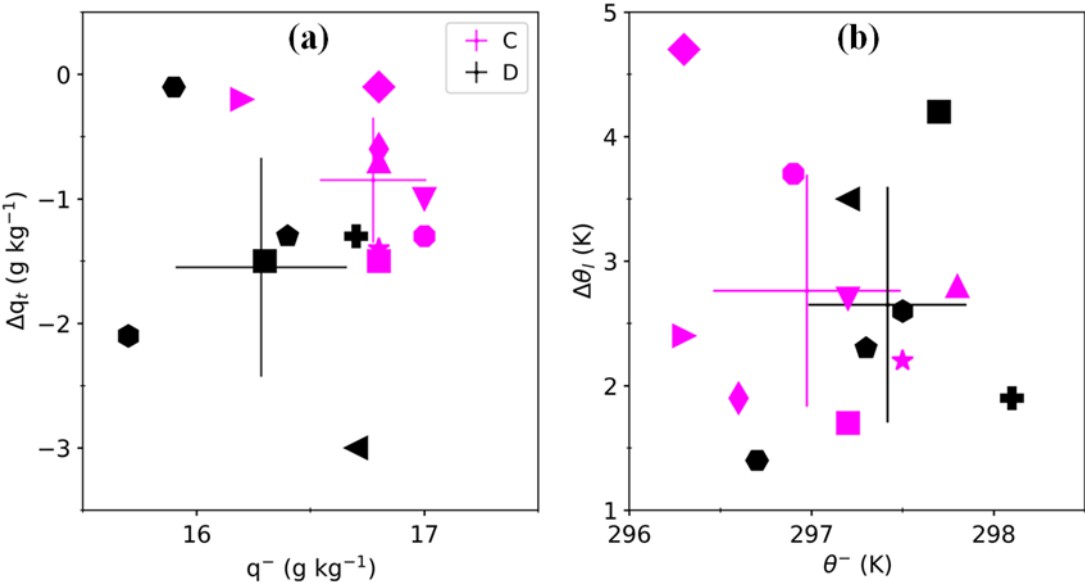

Figure 8: **(a)** Moisture jump at LLSC top ($\Delta q_t$) against specific humidity at the LLSC base ($q^-$), **(b)** temperature jump at LLSC top $\Delta \theta_l$ (possible underestimation of around 1.2 K) against potential temperature at the LLSC base ($\theta^-$), derived from fourteen 05:00 UTC standard morning soundings, for which the probe flew within the LLSC layer (Table A-1). In each panel, the error bars correspond to the standard deviation, and cross at the mean over all C (magenta) or D (black) cases. Each symbol represents a single case.

In C cases, $q^-$ ranges within the interval 16-17 g kg$^{-1}$, with a mean of 16.8 g kg$^{-1}$ and standard deviation of 0.5 g kg$^{-1}$. It is lower in D cases, with an average of 16.3 g kg$^{-1}$ and standard deviation of 0.9 g kg$^{-1}$. Thus, in early morning, the air just below the LLSC is on average 0.5 g kg$^{-1}$ moister in coupled cases. This is qualitatively true for the entire *stratus phase*, when analyzing reusable soundings of the nine IOPs (not shown). $\Delta q_t$ is in absolute overall lower than 3.0 g kg$^{-1}$. It is smaller than or equal to 1.5 g kg$^{-1}$ in 85% of all cases. This indicates a generally weak moisture jump across the LLSC top. This is still

more pronounced in C cases, for which $\Delta q_t$ remains lower than 1.5 g kg$^{-1}$.

    The parameter $\theta^-$ ranges within 296-299 K. Beyond the same variability found in C and D cases, $\theta^-$ is on average around 0.5 K cooler in C cases, probably because the LLSC base is closer to the surface. $\Delta \theta_l$, which varies within the interval 1-5 K, does not exhibit a clear difference between C and D cases. Thus, the fact that the LLSC base gets closer to surface in coupled cases does not impact the temperature jump across the cloud top.

The magnitudes of $\Delta \theta_l$ and $\Delta q_t$ observed in SWA conditions are much smaller than those typically found for the mid-latitude stratocumulus, which can be as strong as 10 K and -10 g kg$^{-1}$ (Duynkerke et al., 2004; Wood, 2012; van der Dussen et al., 2016; Ghonima et al., 2016), especially over the ocean. **The vertical profile used by Pedruzo-Bagazgoitia et al.**

**(2020) to initialize their LES had a $\Delta\theta_l$ of 4.5 K and no jump of $q_t$ across the LLSC top.** This representation is consistent with what we find for the moisture jump, but is on sidelines for the temperature jump.

Table 1: Median and standard deviation of some parameters in the RAD, ENT and SUBS formulation estimated from the fourteen 05:00 UTC radiosoundings presented in Figure 8. The standard deviation (in brackets) over the cases is not indicated when negligible. Our results are compared with the values used in van der Dussen et al. (2014).

| Parameters | Order of magnitude | |
|---|---|---|
| | DACCIWA cases | Study case of van der Dussen et al. (2014) |
| $\overline{T}$ | 294 (0.7) K | 283 K |
| $\overline{q}$ | 16.2 (0.5) g kg$^{-1}$ | 8.2 g kg$^{-1}$ |
| $\rho C_p \Delta F_{rad}$ | 55 (5) W m$^{-2}$ | 48 W m$^{-2}$ |
| $\gamma$ | ~1.012 g kg$^{-1}$ K$^{-1}$ | 0.55 g kg$^{-1}$ K$^{-1}$ |
| $\eta$ | ~ 0.28 | 0.42 |
| $\Gamma_{ql}$ | ~ -2.29 g kg$^{-1}$ km$^{-1}$ | -1.86 g kg$^{-1}$ km$^{-1}$ |
| $w_e$ | 10.12 (2.53) mm s$^{-1}$ | -- |

Table 1 compares our estimates of some parameters involved in the formulation of RAD, ENT and SUBS terms with those of van der Dussen et al. (2014) study case, which are based on the DYCOMS-II (Second Dynamics and Chemistry of Marine Stratocumulus field study) case setup (Stevens et al., 2005). **Our estimates of $\gamma$, $\eta$, and $\Gamma_{ql}$ differ from typical values used by these authors because the LLSC layer for DACCIWA cases is on average 11 K warmer and 8 g kg$^{-1}$ wetter.** After analysis of SBDART model output, $\Delta F_{rad}$ is determined from the difference of net radiative fluxes between model levels just above and below the LLSC layer, respectively. The median and standard deviation of cloud-top longwave radiative cooling are respectively of about 55 and 5 W m$^{-2}$. Our estimate of radiative cooling at the LLSC top for the 25-26 June 2016 case is 44.6 W m$^{-2}$ (Table A-1), which is in good agreement with the value of 43 W m$^{-2}$ estimated by the LES of Pedruzo-Bagazgoitia et al. (2020) for the same day just before sunrise. Despite a weaker temperature and nearly absent moisture jumps at the LLSC top, the median value of our estimated cloud-top radiative cooling is around 10 W m$^{-2}$ greater than that of van der Dussen et al. (2014) and falls within 50-90 W m$^{-2}$, which is the typical interval range for subtropical stratocumulus (Wood, 2012). This is most likely because the LLSC of DACCIWA cases is significantly warmer.

We find only a 5 W m$^{-2}$ standard deviation for radiative cooling at the LLSC top and no significant difference between C and D cases. This very low standard deviation may be due to the conditions, which remained very steady from one case to the other, but may also be underestimated because impacts of higher clouds are not fully included in the estimate of radiative

fluxes. In order to evaluate the error due to temperature underestimation above the LLSC top, SBDART is run with both the measured and a corrected temperature profiles, while the other inputs remain unchanged. The correction of the potential temperature vertical profile consists of a linear tendency between the measured θ plus a 1.2K correction right above CTH and the measured θ at 800 m, where we consider that the radiosonde sensors are no longer affected by the LLSC crossing.

The cloud-top radiative cooling estimated by SBDART with this corrected temperature vertical profile is larger by less than 2 W m$^{-2}$.

The cloud-top entrainment velocity, $w_e$ (Eq. 3), has a median value of 10.12 mm s$^{-1}$ and its variability is around 25% of the median. This median is around 2.5 times higher than the velocity obtained by Pedruzo-Bagazgoitia et al. (2020) with LES and among the highest values found by other authors (Duynkerke et al., 2004; Faloona et al., 2005; Mechem et al.,

2010; Ghonima et al., 2016). Finally, we show that our estimates of RAD and ENT terms are suitable, beyond potential errors on the entrainment efficiency $A$, and simplified settings in SBDART. As mentioned in section 3.3, we approximate the SUBS term with the assumption of a stationary LLSC top at sounding time (Eq. 6). This term must be taken with more caution than the other two, due to this hypothesis.

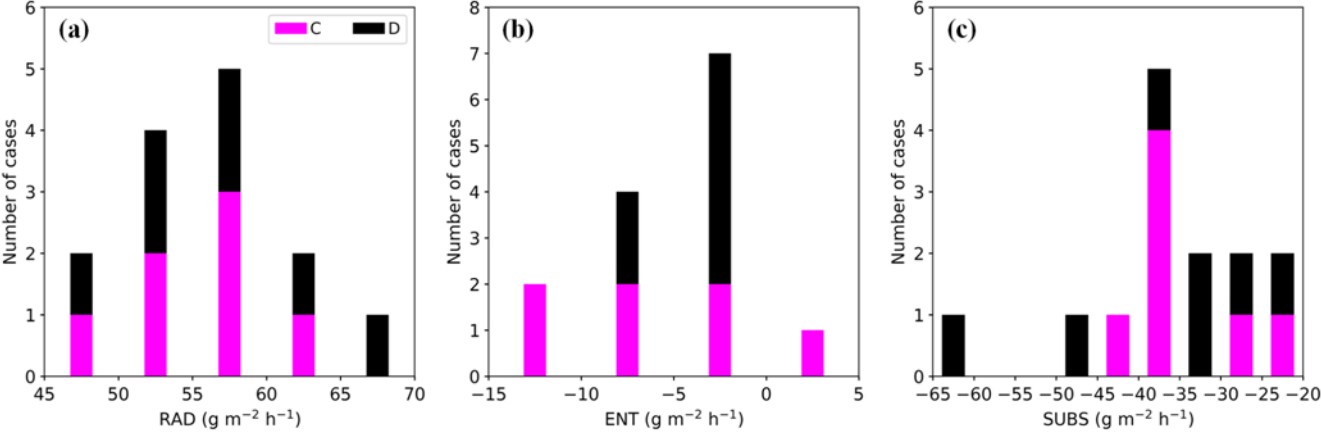

Figure 9: Distributions of radiative (RAD, a), entrainment (ENT, b) and large-scale subsidence (SUBS c) LWP budget terms (Eq. 1), derived from the fourteen 05:00 UTC standard soundings at Savè supersite for which the probe crossed into the LLSC layer (Fig. 8 and Table A-1). The methodology is described in section 3.3.

Figure 9 presents distributions of RAD (Fig. 9a), ENT (Fig. 9b) and SUBS (Fig. 9c) terms derived from the fourteen

radiosoundings considered in Fig. 8 by the methodology described in section 3.3. The RAD term ranges within 45-70 g m$^{-2}$ h$^{-1}$, with a median of 57 g m$^{-2}$ h$^{-1}$. ENT varies between -15 and 5 g m$^{-2}$ h$^{-1}$, indicating a smaller contribution to the LWP budget compared to RAD. The negative value of about -10 g m$^{-2}$ h$^{-1}$ is consistent with the study of Pedruzo-Bagazgoitia et al.

(2020), with a predominant role of cloud-top temperature and moisture jumps and a drying and warming entrainment effect. Among the fourteen cases, several have a smaller ENT contribution than this. One case even has a positive value for ENT, which means that the LLSC depth has more impact than temperature and moisture jumps, so that the entrainment in that case favours LLSC deepening. The SUBS term ranges between -65 and -20 g m$^{-2}$ h$^{-1}$, with a median of around -36 g m$^{-2}$ h$^{-1}$. It corresponds to as much as -0.4 to -0.9 times the RAD term, which is very significant. This is also consistent with Pedruzo-Bagazgoitia et al. (2020), who found a SUBS/RAD ratio of approximately -0.4 before sunrise. Our answers to the two questions raised at the start of this sub-section are:

1) We found similar results compared to Pedruzo-Bagazgoitia et al. (2020). However, the West African inland LLSC layer, which develops within the monsoon flow (Dione et al., 2019), is characterized by weaker temperature and moisture jumps, but with similar radiative cooling at its top compared to marine stratiform clouds.

2) The cloud-top radiative cooling and the three LWP budget terms RAD, ENT and SUBS do not exhibit significant differences between the C and D cases, because of similar cloud depth and thermodynamic characteristics. The slight differences in CBH and moisture jump across the cloud top between the two types of cases do not impact cloud-top radiative cooling and LWP budget analysis at the end of the *stratus phase*.

Through a series of sensitivity tests based on horizontal wind speed profiles, Pedruzo-Bagazgoitia et al. (2020) found that wind shear at the LLSC top before sunrise, such as observed during the DACCIWA experiment (Lohou et al., 2020), may accelerate the cloud deck breakup during the *convective phase* by generating dynamical turbulence which enhances the ENT term. However, they did not investigate the effect of wind shear below the LLSC.

From the fourteen morning soundings considered in Fig. 8, we quantified the contribution of vertical shear to the production of turbulence at the LLSC top (Table A-1). We find it to be generally smaller than $20 \times 10^{-5}$ s$^{-2}$; that is, considerably smaller than that imposed at the initialization of LES experiments performed by Pedruzo-Bagazgoitia et al. (2020). However, this contribution in the subcloud layer is mostly higher than $50 \times 10^{-5}$ s$^{-2}$ (Fig. 4c). Thus, the dynamical instability induced by the NLLJ is more important below the LLSC layer than above. This should imply that the mechanical turbulence driven by the NLLJ impacts the turbulent fluxes at LLSC base much more than entrainment of ambient air from above.

## 4.3 Factors controlling the coupling

Previous studies have demonstrated that several processes may lower the LLSC base and couple the cloud deck with the surface during the *stratus phase*: (i) shear-driven turbulence in the subcloud layer (Adler et al., 2019; Babić et al., 2019a), (ii) cloud droplets sedimentation at the cloud base (Dearden et al., 2018), (iii) light precipitation formation (i.e. drizzle) in the subcloud layer (Wood, 2012), (iv) convective overturning driven by the cloud-top radiative cooling (Wood, 2012), and, (v) large-scale advection (Zheng and Li, 2019). Sections 4.1 and 4.2 allowed us to test several of these hypotheses to understand why the LLSC couples to the surface in some DACCIWA cases.

As discussed in section 4.1, there is no difference in shear-driven turbulence between C and D cases, which could explain the thermally neutral stratification of the subcloud layer in C cases and the stable stratification in D cases. Therefore, the NLLJ does not appear to be responsible for the LLSC coupling in C cases.

With LES experiments based on the 04-05 July case (case D, IOP7), Dearden et al. (2018) hypothesized that the LLSC base descent during night is due to cloud droplets sedimentation at the cloud base. However, the cloud base decrease is of less than 50 m before sunrise in this numerical experiment, whereas the observed LLSC base descent is larger than 100 m by the end of the *stratus phase* in most of our studied cases, either C or D. Thus, cloud droplets sedimentation alone cannot explain the coupling in C cases.

In all the DACCIWA cases we study, no precipitation was recorded at the surface during the *stratus phase*. However, drizzle formation below the LLSC base can hardly be measured by rain-gauge sensors. Therefore, this hypothesis cannot be fully tested and remains a possibility. In terms of radiative cooling at the LLSC top, section 4.2 shows that this positive contribution to the LWP budget at the end of the *stratus phase* is similar in the C and D cases.

The large-scale effects must be considered not only in the LLSC formation (Babić et al., 2019b), but also in its diurnal cycle. Indeed, eight of the nine C cases are observed between 26 June and 8 July 2016 (Table A-1). This period corresponds to the first days of post-onset phase characterized by a well-established and undisturbed monsoon flow over SWA (Knippertz et al., 2017). Warm air advection was observed to decouple LLSC layer from the surface (Zheng and Li, 2019). Therefore, the reverse process, i.e. colder air advection, may produce the opposite effect. This hypothesis is all the more likely since LLSC formation during the West African monsoon season is mainly due to horizontal advection of cooler air. The reusable soundings performed during the *stratus phase* of the nine IOPs revealed that, at 50 m a.g.l (sounding level below the lowest CBH at the end of *stratus phase*) the relative humidity remains larger than 90 % for all the cases (not shown). For C cases, a decrease in specific humidity (by around 1 g kg$^{-1}$) and a slight decrease in temperature (by around 0.2 °C) are observed between LLSC formation and its coupling with the surface, which maintains constant Rh. However, no clear tendency was observed in D cases. The very small tendency of temperature and humidity and the small number of studied cases do not allow us to definitively conclude an effect of cooling and drying due to the horizontal advection of maritime air. However, this advection seems to persist in C cases and could have some impact, though not on LLSC base lowering (because Rh is constant at 50 m a.g.l); rather, the dry advection may have an effect on the LCL evolution. Indeed, a 1 g kg$^{-1}$ decrease of near-surface specific humidity implies an elevation of surface-based LCL by a hundred meters, which facilitates the coupling.

In summary, none of processes listed at the beginning of this sub-section is solely responsible for the coupling before sunrise. We can hypothesize that it is combination of several of those processes, each with a small impact that leads to LLSC layer coupling with the surface. After the coupling, turbulence underneath the LLSC plays a crucial role in its maintenance during the rest of the *stratus phase*, as indicated by the reduction of thermal stability in the subcloud layer for C cases (Fig. 5b). Indeed, the contributions of shear-driven turbulence below the NLLJ and convective turbulence due to the cloud-top radiative cooling are important for mixing potential temperature in the subcloud layer (Dione et al., 2019; Lohou et al.,

2020). In LES experiments under windless conditions carried out by Pedruzo-Bagazgoitia et al. (2020), cloud-top radiative cooling was the sole source of turbulence in the ABL until sunrise, and the coupling between cloud and surface was maintained.

**5 Evolution of the LLSC layer under daytime conditions**

In this section, the *convective phase* of the LLSC diurnal cycle is analyzed.

**5.1 The three scenarios of evolution**

The LLSC evolution during the *convective phase* is first analyzed according to ceilometer-derived CBHs temporal change

10 relatively to surface-based LCLs. From this point of view, all C cases evolve quite similarly during this phase (scenario C), while two distinct scenarios are observed among D cases (hereafter named DC for "decoupled-coupled" and DD for "decoupled-decoupled"). Each of the three scenarios is illustrated by one typical example: the LLSC occurrence on 07-08 July (Fig. 10a) for scenario C, 25-26 June (Fig. 10b) and 04-05 July (Fig. 10c) for scenarios DC and DD, respectively.

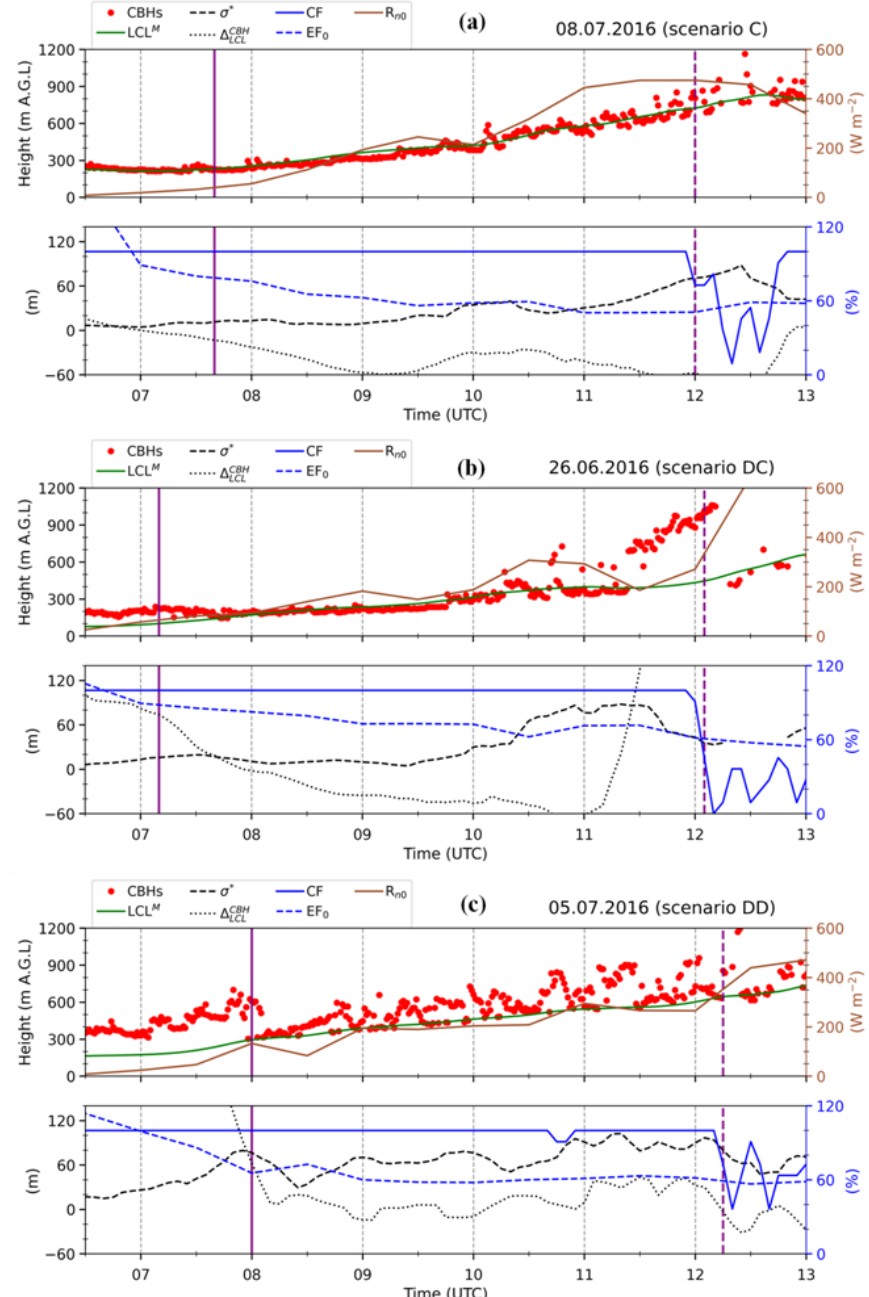

Figure 10: Illustration of the three scenarios of LLSC evolution after sunrise observed at the Savè supersite during DACCIWA field campaign: **(a)** 08 July 2016 for scenario C, **(b)** 26 June 2016 for scenario DC and **(c)** 05 July 2016 for scenario DD. The top panels present ceilometer-derived CBHs, lifting condensation level (LCL) and net radiation measured at surface ($Rn_0$). The bottom panels gather cloud fraction (CF), evaporative fraction at surface ($EF_0$ in %), standard deviation of the cloud base height in LLSC layer ($\sigma^*$) and mean distance between cloud base height and surface-based LCL ($\Delta_{LCL}^{CBH}$). The vertical solid and dashed lines indicate the surface-convection influence time ($T_i$) and the LLSC deck breakup time ($T_b$), respectively. The Local time at Savè (Benin) is UTC +1 hour.

Whether the CBHs is close to LCL (Fig. 10a) or not (Fig. 10b and c), it has a low variability before 07:00 UTC in these three illustrative cases, indicating a quite horizontally homogenous base of the LLSC layer before the start of the *convective phase* (as seen in the previous section). The CBHs and LCL in scenario C lift together after 07:30 UTC due to thermal convective conditions in the subcloud layer. After 09:00 UTC, σ* increases gradually, but the lower bases always fit with

LCL, with $\Delta_{LCL}^{CBH}$ ranging between 0 and -40 m (Fig. 10a, lower panel). This can be interpreted as a progressive change in the LLSC base structure, which is more and more heterogeneous in height, but the cloud layer remains coupled with the surface all along. The evolution from stratus to stratocumulus and eventually to cumulus cannot be established using CBH alone, but the ceilometer-derived CBHs already show a clear evolution from homogeneous LLSC towards a more heterogeneous low cloud structure until the cloud deck breakup time, established when CF decreases to less than 90 %, which happens at 12:00

UTC on 08 July 2016.

The LLSC in scenario DC (Fig. 10b) is decoupled from the surface at the end of the *stratus phase*. The LCL starts to rise at 07:00 UTC and joins the LLSC base about 1 hour later, indicated by a decrease of $\Delta_{LCL}^{CBH}$ down to zero (Fig. 10b, lower panel). After the coupling, scenario DC is very similar to scenario C and will be discussed further in section 5.3.

The LLSC evolution in scenario DD (Fig. 10c) is quite different from the other two. The LLSC layer remains decoupled

from the surface until 08:00 UTC, as shown by a significant departure between CBHs and LCL ($\Delta_{LCL}^{CBH} > 120\ m$, Fig. 10c, lower panel), due to a similar lifting rate of both levels. After 08:00 UTC, a new cloud layer with a base very close to LCL ($\Delta_{LCL}^{CBH} < 40\ m$), is detected 200 m below the LLSC deck. The values of σ*, much larger than 60 m after 08:30 UTC, indicate that this new cloud layer rapidly turns to shallow cumulus clouds. Unfortunately, it is not possible to distinguish both cloud layers with ceilometer-derived CBHs, because they remain too close together, with variable cloud bases and edges.

However, we can assume that the LLSC layer formed during the night remains above the cumulus clouds during part of the *convective phase*. The higher CBHs detected by the ceilometer after 09:00 UTC are the overlying LLSC base (about 200 m higher). The cumulus and LLSC layers above can, however, clearly be seen on visible and infra-red full sky cameras (not shown). In the case where the two cloud layers are superimposed, two possibilities may occur: (i) the underlying surface-convection-driven cumulus clouds do not interact with LLSC deck, which remains decoupled from the surface, (ii) the

underlying cumulus clouds develop vertically, reach the LLSC layer, and act to intermittently and locally couple it with the surface (Wood, 2012).

Among the thirteen D cases observed at the end of the *stratus phase*, eight and five follow scenario DD and DC, respectively, during the *convective phase* (Table A-1). The main difference between the three scenarios is that the first shallow convective clouds form when the LLSC layer breaks up in scenarios C and DC, whereas in scenario DD, shallow

cumulus clouds form below the LLSC deck before it breaks up. Similar transitions were reported by previous observational and modelling studies on the stratiform low-level clouds (Price, 1999; Xiao et al., 2011; Ghonima et al., 2016; Mohrmann et al., 2019; Sarkar et al., 2019; Zheng and Li, 2019; Pedruzo-Bagazgoitia et al., 2020). **In particular, the Sc-Cu transition of scenario DD is part of the conceptual model for marine stratocumulus (Xiao et al., 2011; Wood, 2012).**

What conditions lead the LLSC to either be coupled to the surface in scenario DC, or to remain possibly decoupled with the formation of an underlying cumulus cloud layer in scenario DD? No relevant differences in macrophysical characteristics of LLSC (base and depth) were found between the two scenarios at the end of the *stratus phase* and beginning of the *convective phase* (not shown). The LLSC with low bases are not systematically those which will be coupled to surface at the

5 beginning of *convective phase*. The four parameters presented in Fig. 8, which summarize thermodynamical conditions below and above the LLSC layer, are not fundamentally different between the DC and DD scenarios either. The relative humidity in the subcloud layer at the end of the *stratus phase* is larger than 95 % in all D cases, and the difference between scenarios DD and DC is smaller than 2 %, which is about the measurement accuracy. Consequently, alternative approaches are needed to identify the processes involved in the LLSC coupling with surface during the *convective phase*.

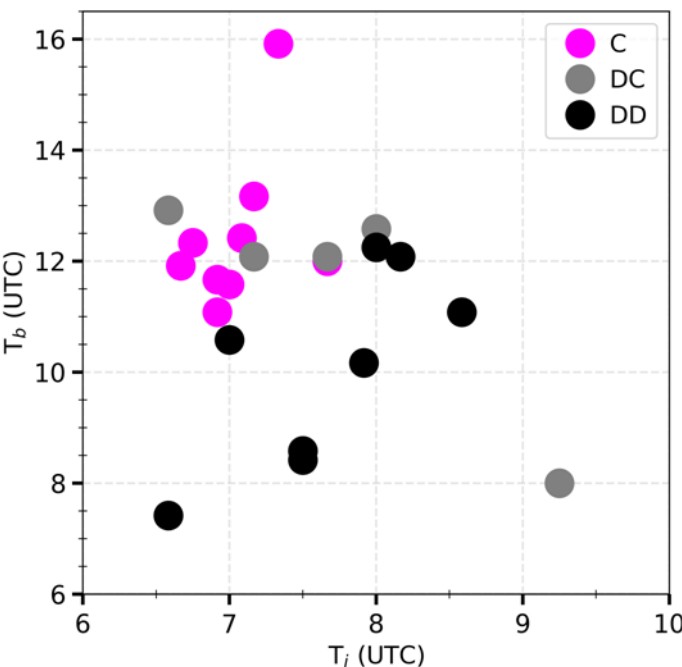

Figure 11: LLSC breakup time ($T_b$) against surface-convection influence time ($T_i$) for the twenty-two selected cases (Table A-1). The colors represent the three different scenarios.

In conclusion, the coupling between LLSC layer and surface during the *convective phase* appears to be the key factor in determining how the transition towards shallow convective clouds takes place. When the LLSC is coupled to the surface (C and DC cases), it is the breakup of the cloud deck that leads to the formation of different low-level clouds types

(stratocumulus or cumulus). When the LLSC is decoupled from the surface (DD cases), the shallow convective clouds form below it. In the next sub-section, we analyze the different scenarios of LLSC evolution in greater depth.

### 5.2 Surface-convection and breakup times

**We defined two characteristic times of the LLC evolution (see section 3.2): the surface-convection influence and LLSC breakup times ($T_i$ and $T_b$, respectively). $T_b$ is determined by the diagnostic parameter CF. $T_i$, which indicates when the low cloud coverage is influenced by the surface-buoyancy-driven turbulence, is defined differently according to the scenario.** For scenario C, $T_i$ corresponds to the time when the LLSC base starts to lift together with LCL. After sensitivity tests, $T_i$ is defined as the first time when $LCL^M$ increases to at least 5 m above its value at 06:30 UTC. For scenario DC, $T_i$ corresponds to the time when the rising LCL reaches LLSC base; that is, when the LLSC layer is coupled to the surface ($\Delta_{LCL}^{CBH} < 75$ m, which is also the threshold used to differentiate C and D cases at the end of the *stratus phase* in section 4.1). For scenario DD, $T_i$ is the first time when new low clouds appear below the LLSC deck. As these clouds are coupled to the surface, $T_i$ is also determined when $\Delta_{LCL}^{CBH}$ decreases to less than 75 m.

Figure 11 displays $T_b$ and $T_i$ for the twenty-two LLSC cases (Table A-1). $T_i$ ranges between 06:30 and 09:15 UTC. $T_b$ varies between 07:30 and 16:00 UTC, with breakup time occurring before 12:00 UTC in 72% of cases. The latter result is consistent with the findings of Dione et al. (2019), who used infrared sky camera images to define the LLSC lifetime. We can see that the LLSC breakup time is not linked to the time at which it starts to rise or at which underlying cumulus clouds form.

For scenario C, $T_i$ hardly changes from one case to the other. It ranges between 06:40 and 08:00 UTC, which is not long after sunrise (06:00 UTC). The LLSC persists for at least 4.5 hours and breaks up between 11:00 and 16:00 UTC. The latest breakup time, occurring at 16:00 UTC, corresponds to the 02-03 July 2016 case, for which the collocated radar reveals light precipitation from higher clouds (above LLSC layer) during the first hours of the *convective phase* (not shown), while nothing was recorded by the surface rain-gauge. This external forcing, able to enhance the liquid water content in the LLSC layer, is certainly responsible for this late breakup. Because this case is an exception and cannot easily be compared to the others, it is not considered hereafter.

For four out of five DC cases, $T_i$ and $T_b$ are very close to values observed for C cases. This means that the stable stratification in the subcloud layer before the *convective phase* (which allowed classification of this case as decoupled during the *stratus phase*) is rapidly eroded after sunrise and does not seem to impact the breakup time. The case for which $T_b$ occurred at 08:00 UTC (16-17 July 2016) is removed in the following as well, because the LLSC breaks up before LCL reaches its base.

The DD scenario presents the largest variation ranges of $T_i$ (between 06:35 UTC and 09:00 UTC) and $T_b$ (between 07:00 UTC and 13:00 UTC). The most striking result is that the LLSC in scenario DD often breaks up earlier than in scenarios C and DC.

Following the LES of Pedruzo-Bagazgoitia et al. (2020), the start of the *convective phase* leads to three main changes in the LWP tendency equation. First, the radiative cooling (RAD term) decreases due to solar heating at the cloud top. Second, the ENT term also strongly decreases because the thermally-driven convection enhances entrainment of dry and warm air from aloft into the LLSC layer. Third, the BASE term, which was close to zero during the *stratus phase*, comes into play during the *convective phase* and contributes positively to $\frac{\partial LWP}{\partial t}$. Despite the BASE term, the strong decrease of both ENT and RAD makes $\frac{\partial LWP}{\partial t}$ negative one hour after sunrise. The RAD and ENT terms cannot be estimated during the *convective phase* with the dataset acquired at Savè because several data are missing, among them the CTH.

The C and DC scenarios during the convective phase are very close to the case simulated in Pedruzo-Bagazgoitia et al. (2020) and we can expect a quite similar evolution of terms involved in the LWP prognostic equation. Conversely, the DD scenario might be very different. The LLSC breaks up earlier, mostly before or around 10:30 UTC, when it is decoupled from the surface, likely due to a weaker BASE term. This hypothesis is supported by the findings of van der Dussen et al. (2014) suggesting that LLSC coupled to the surface moisture are more resistant to cloud-thinning related processes, such as the entrainment of dry and warm air into the cloudy layer. The stronger variability of breakup time for DD cases may come from the fact that the LLSC thinning depends on its interaction with the underlying cumulus clouds. If the latter penetrate the LLSC deck, local coupling can happen, which induces a homogeneous cloud layer from the surface to LLSC top, but, at the same time, entrainment at the cloud top is enhanced by the vertical development of cumulus (Wang and Lenschow, 1995).

The LLSC breakup time impacts the surface radiative budget over the day, then the surface fluxes, and consequently, the vertical development of ABL, as shown by Lohou et al. (2020). They estimated that the ABL height is about 900 m when the LLSC deck breaks up at 09:00 UTC and is 30% lower when this breakup occurs at 12:00 UTC. Consequently, one can expect a quite different vertical development of ABL in C/DC cases compared to DD cases.

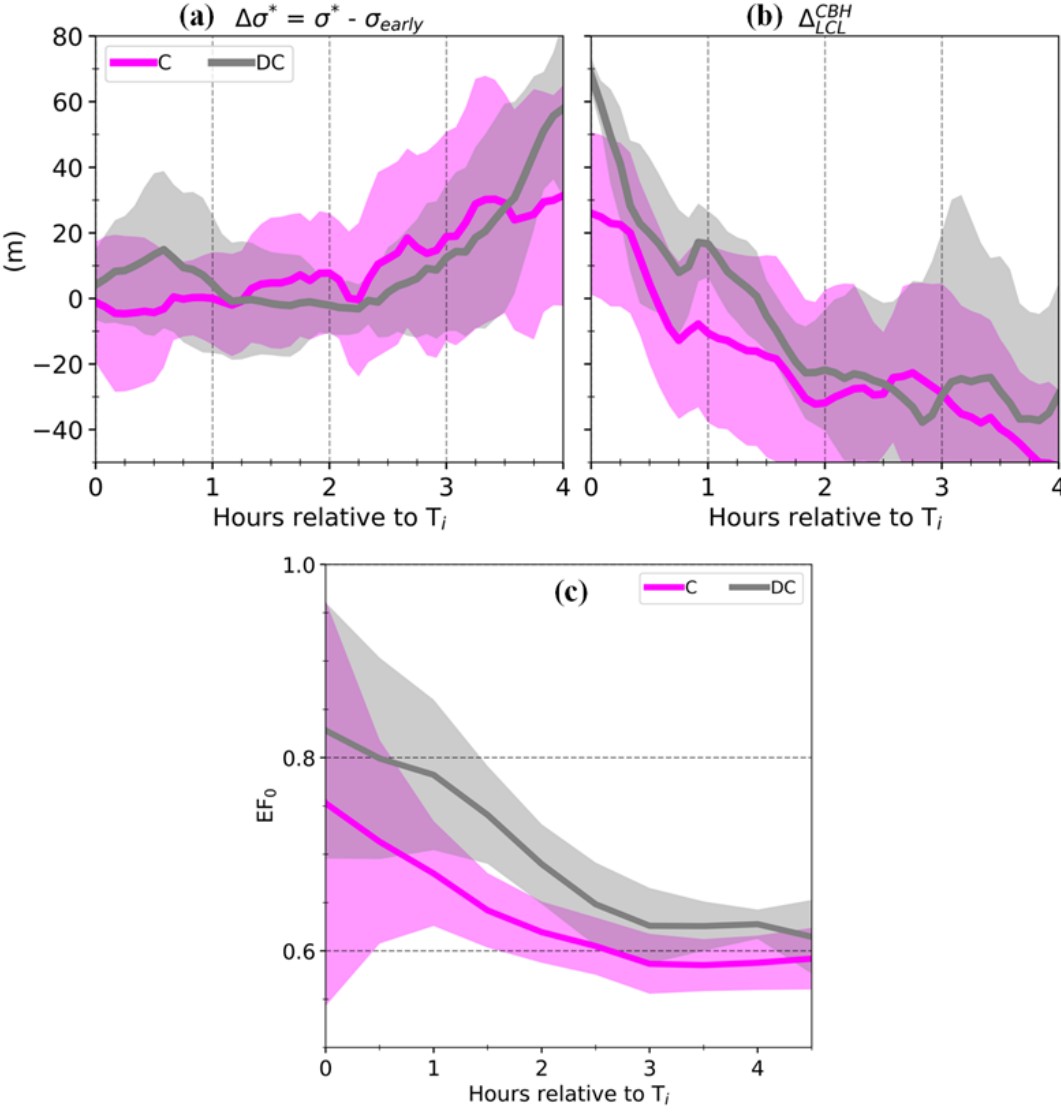

Figure 12: Evolution of, **(a)** Δσ*, which is the difference between the diagnostic parameter σ* and its median over the period from 04:00 to 06:30 UTC on day-D+1 ($\sigma_{Early}$), **(b)** the mean distance between the LLSC base height and surface-based LCL ($\Delta_{LCL}^{CBH}$), **(c)** the evaporative fraction at surface ($EF_0$), for C (coupled) and DC (decoupled-coupled) scenarios. The solid lines indicate the median and shaded areas represent the standard deviation. The time is expressed in hours relative to surface-convection influence time ($T_i$).

### 5.3 Evolution of the LLSC horizontal structure for C and DC cases

The changes in the LLSC horizontal structure for C and DC scenarios are now further analyzed based on the evolution of the LLSC base and its standard deviation $\sigma^*$. The DD cases are excluded from this analysis because the macrophysical characteristics of associated LLSC cannot be determined after the underlying clouds formation. As illustrated in Fig. 10a and b, the elevation rate of LCL, and consequently of LLSC base, may change a lot from one case to the other. It is about 108 m $h^{-1}$ and 67 m $h^{-1}$ for 8 July and 26 June, respectively. It could be expected that the higher this rate, the higher $R_{n0}$, and the more intense the thermally-driven convection in the subcloud layer as well as the corresponding BASE term. However, no clear link is pointed out between $T_b$ and this elevation rate of LLSC base (not shown).

Contrary to LLSC base height, $\sigma^*$ has a common tendency among all the C and DC cases. The evolution of $\sigma^*$ with time compared to its value at $T_i$, $\sigma_{Early}$, is presented in Fig. 12a. A four-hour period is considered here because it is the smallest duration between $T_i$ and $T_b$ for the twelve C and DC cases included in this statistic (Fig. 11). As also illustrated in Fig. 10a and Fig. 10b, $\sigma^*$ remains close to $\sigma_{Early}$ for at least two hours after $T_i$ (until 09:00 UTC for 8 July and 09:30 UTC for 26 July). Consequently, during this period, the structure of LLSC bases remains quasi-unchanged. Afterwards, $\sigma^*$ progressively increases for at least 2 hours until the LLSC deck breakup. From $T_i$ to the breakup, $\Delta_{LCL}^{CBH}$ remains lower than 70 m, with even a slight decrease in the first two hours (Fig. 12b), suggesting an enhancement of coupling due to the increase of thermally-driven turbulence in the subcloud layer. The combination of (1) very heterogeneous LLSC base and (2) the fact that the lowest cloud bases remain close to LCL during the few hours before $T_b$, indicates that some of the bases are coupled to the surface but some tend to be decoupled from the surface.

Eventually, the evolution of $\sigma^*$ and $\Delta_{LCL}^{CBH}$ (Fig. 12) allows two periods to be defined between $T_i$ and $T_b$: (1) the first two hours after $T_i$, during which the LLSC deck is fully coupled to the surface and the homogeneity of its base is not yet affected and, (2) the few hours before $T_b$ during which the base of LLSC layer becomes more and more heterogeneous and intermittently decoupled from the surface. This latter tendency can be seen in Fig. 10a and b (upper and lower panels) after 11:00 UTC and 10:15 UTC, respectively. A decoupling of the LLSC layer from the surface is also observed about half an hour before its breakup time in the LES of Pedruzo-Bagazgoitia et al. (2020).

The bottom panels of Fig. 10 present the evolution of evaporative fraction at the surface ($EF_0$) for the illustrative cases. Figure 12c displays the medians of this parameter over all C and DC cases. Defined as the ratio of $LHF_0$ to ($LHF_0 + SHF_0$), an $EF_0$ larger than 0.5 means that evapo-transpiration dominates over warming. This was on average the case at Savè during the DACCIWA campaign (Kalthoff et al., 2018). Figure 12c shows that the median of $EF_0$ decreases from around 0.75 at $T_i$ to 0.6 at LLSC breakup. The predominance of evapo-transpiration over sensible heat flux, particularly during the first two hours after $T_i$, and the full LLSC coupling to the surface, might contribute to maintaining this cloud layer throughout the BASE term. The LLSC base is indeed strongly homogeneous. The decrease of $EF_0$ and its levelling at 0.6 implies a faster

increase of SHF$_0$ than LHF$_0$. We can then expect a larger contribution of $\overline{w'\theta_l'}^b$ and a smaller one from $\overline{w'q_t'}^b$ in the BASE term with time. This favours the convection in the LLSC layer, which enhances cloud top entrainment, at the expense of cloud moistening by underlying turbulent mixing. In addition to this, the final intermittent decoupling of LLSC layer from the surface likely contributes, together with the decrease of RAD and ENT terms (Pedruzo-Bagazgoitia et al., 2020), to the breakup of the cloud deck.

It appears that the LLSC and timing of its evolution in scenarios C and DC are very similar during the *convective phase*. In these scenarios, the LLSC keeps the same characteristics in terms of coupling and base homogeneity for two hours after T$_i$. Afterwards and until its breakup, the LLSC becomes more and more heterogeneous and intermittently decoupled from the surface. These two steps are in phase with the evolution of EF$_0$ that likely impacts the BASE term, which is the only positive contribution to the LWP budget during the *convective phase*.

## 6 Summary and conclusion

**The objective of this study is to examine the breakup of almost daily LLSC during the monsoon season in southern West Africa.** It is based on the analysis of a set of twenty-two precipitation-free LLSC occurrences observed at the Savè supersite during the DACCIWA field experiment. The diurnal cycle of the LLSC consists of four main stages and this study addresses the last two, the *stratus* and *convective* phases. We used the ground-based observational data collected by (i) ceilometer and cloud radar for the cloud layer macrophysical properties, (ii) energy balance and weather stations for atmospheric conditions near the surface, and finally, (iii) radiosoundings and UHF wind profiler for thermodynamical and dynamical conditions within the low-troposphere. From these measurements, some diagnostics of the LLSC layer are estimated, including: cloud base height, cloud coverage fraction, cloud base homogeneity and cloud layer coupling with the surface. The coupling was assessed by the distance between the LLSC base height and surface-based lifting condensation level; the cloud layer is coupled to the surface when these two levels coincide. Our main results are summarized in Fig. 13 by a schematic illustration.

At the beginning of the *stratus phase* (after 22:00 UTC), the LLSC is decoupled from the surface in all but one of the studied cases. Over the following four hours, in nine of the twenty-two cases, the LLSC base lowers in such way that the cloud layer becomes coupled to the surface (referenced as cases C, Fig. 13c). In the other thirteen cases (referenced as cases D, Fig. 13a and b), the LLSC remains decoupled from the surface. The weak thermodynamical differences observed between the C and D cases at Savè cannot fully explain the coupling which occurs in C cases. However, the C cases occurred preferentially between 27 June and 8 July 2016, a period with a well-established monsoon flow over West Africa, especially over the DACCIWA investigated area. Most of the D cases are observed during the monsoon-onset period or during disturbed sub-periods after 08 July 2016. If the synoptic conditions of monsoon flow play a role in the LLSC coupling with the surface, it could be through thermodynamical conditions, which were only slightly apparent in the Savè dataset. It could also be through large-scale dynamical parameters like large-scale subsidence, which is an important factor in the LWP

budget and could not be determined precisely for every day with the Savè dataset. The analyses of stable and jet phases by Adler et al. (2019) and Babić et al. (2019a,b) outline a complex imbrications of different processes in LLSC formation. Similarly, we conclude that the LLSC coupling to the surface during the *stratus phase* is also based on different processes for which a slight intensity change may have an important impact.

The Savè dataset allowed us to estimate the most important terms of the LWP tendency equation at the end of the *stratus phase*, notably the radiative, entrainment and subsidence terms. Our values are very close to those found by Pedruzo-Bagazgoitia et al. (2020) in a numerical study of a DACCIWA case. Since the LLSC layer develops in the monsoon flow, it is warmer and characterised by weaker temperature and humidity jumps at its top, but with the same magnitude order of cloud-top radiative cooling, compared to marine stratocumulus over the subtropical region.

During the *convective phase* of the LLSC diurnal cycle, a new separation occurs among D cases. In some, the LLSC couples to the surface while the lifting condensation level rises with thermally-driven convection at the surface (Fig. 13b). Therefore, the LLSC deck may follow three scenarios until its breakup: (1) scenario DD for "decoupled-decoupled" (followed by most of D cases, Fig. 13a), (2) scenario DC for "decoupled-coupled" (followed the other D cases, Fig. 13b), and (3) scenario C (followed by all C cases of the *stratus phase*, Fig. 13c). Scenarios C and DD are the most frequent among

the twenty-two studied cases, with nine and eight occurrences, respectively. The reason why D cases follow DC or DD was not clearly identified.

       Typically, scenarios C and DC are quite similar and consist of two steps: (ii) the two first hours, during which the LLSC layer lifts but remains fully coupled to surface and homogeneity of its base is not yet affected, (ii) the few hours preceding the breakup time, during which the cloud layer is sometime decoupled from the surface as its base becomes more and more

heterogeneous. In these two scenarios, the breakup of the LLSC deck leads to a transition towards shallow cumulus clouds. This occurs at around 11:00 UTC or later, approximately 4.5 hours after the LLSC starts to lift. In scenario DD, cumulus clouds, triggered by the convectively mixed layer, form below the LLSC deck before its breakup. The breakup time in this scenario varies strongly between 07:30 UTC and noon, but occurs in most cases before 11:00 UTC. The earlier breakup occurring in scenario DD outlines the importance of coupling with the surface for LLSC maintenance after sunrise. Thus, we

conclude that, in SWA conditions, the coupling between LLSC and surface is a key factor for its evolution during daylight hours. It determines the LLSC lifetime and the way in which the transition towards shallow convective clouds occurs. The coupled LLSC last longer (breakup time at 12:00 UTC in average) than decoupled cases (breakup time at 10:00 UTC in average). According to Lohou et al. (2020), this difference in breakup time leads to a reduction of about 15% of net radiation at the surface and of ABL vertical development during the day in coupled versus decoupled cases.

From these results, it appears important to correctly simulate the coupling of nocturnal LLSC layer for a better representation of the West African monsoon features in global climate and weather model simulations. However, the processes responsible for the coupling at different stages of the LLSC diurnal cycle (during the *stratus phase* for C cases (Fig. 13c) and the *convective phase* for DC scenario (Fig. 13b)) are not easy to identify. The coupling results from a combination of several processes rather than a single distinct predominant one. Thus, it is very difficult to recommend one

single improvement in the models. The aerosol loading in the low-troposphere is a potential factor in controlling LLSC evolution and lifetime (Deetz et al., 2018; Mohrmann et al., 2019; Redemann et al., 2020). The airborne measurements of low-cloud properties over SWA during the DACCIWA campaign (Flamant et al., 2017) could be used to assess the microphysical role of aerosol in the LLSC evolution scenario. This may help to differentiate between the DC and DD scenarios. Furthermore, the potentially large influence of middle-level clouds on LLSC also remains an open question and was not objectively addressed in this study. It would be also interesting to study how the LLSC breakup over SWA might change in the future climate.

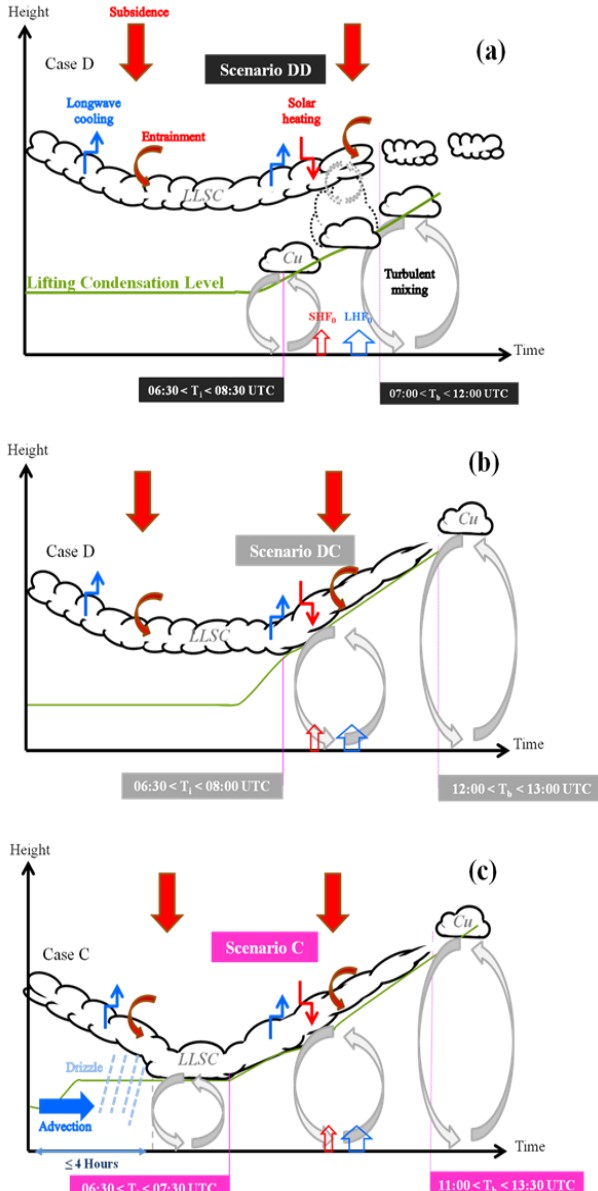

Figure 13: Schematic illustration of the main findings of this present study. It portrays the typical evolutions of LLSC layer sampled at Savè (Benin where local time equals UTC +1 hour), during the DACCIWA field experiment. The different scenarios and their characteristic times as well as the relevant physical processes are illustrated (the meaning of the different arrows is indicated in **a,** and remains the same in **b** and **c**). The representation encompasses stratus and convective phases of the LLSC diurnal cycle. The width of arrows representing the near-surface latent and sensible heat fluxes (LHF$_0$ and SHF$_0$ resp.) correspond to their relative proportions. Typically, the LLSC are decoupled from the surface at formation (**a**, **b** and **c**). For D cases (**a** and **b**), the LLSC remains uncoupled all along the stratus phase. For C cases (**c**), the LLSC gets coupled to surface within the four hours after its formation as cloud base descents significantly and LCL increases, potentially because of drier and cooler air horizontal advection (horizontal blue filled arrow in **c**), and drizzle formation in the subcloud layer (**c**). In all C cases, the LLSC evolves by scenario C, in which the cloud layer lifts with the growing convective boundary layer, the subsequent cloud deck breakup leads to shallow convective clouds formation. In scenario DD (**a**), followed by most of D cases, surface-convection-driven cumulus forms below the LLSC deck before its breakup. The others D cases evolve by scenario DC (**b**), in which the LLSC couples with surface as the convective boundary layer top joins the LLSC base, and the subsequent LLSC evolution is similar to scenario C.

*Data availability*. The data used in this study are available in the BAOBAB (Base Afrique de l'Ouest Beyond AMMA Base) database (https://baobab.sedoo.fr/DACCIWA/).

5  *Author contributions*. FL, NK, ML, CD, BA, XPB and SD performed the measurements at the Savè supersite. MZ processed the data and carried out the analysis with contributions from FL and ML. MZ wrote the paper with contributions from all co-authors.

*Competing interests*. The authors declare that they have no conflicts of interest.

*Acknowledgements*. The DACCIWA project has received funding from the European Union Seventh Framework Programme (FP7/2007-2013) under grant agreement no. 603502. The first author thanks the Laboratoire d'Aérologie, Université de Toulouse, France for hosting the research activities. We would also like to thank two anonymous reviewers and the independent editor Debra Bellon for their helpful comments and suggestions.

*Financial support*. This study received financial support from the PASMU (Pollution de l'Air et Santé dans les Milieux Urbains de Côte d'Ivoire) project funded by the programme of Debt Reduction-Development Contracts (C2Ds) managed by the Institute of Research and Development (IRD, France).

**Appendix A** : LLSC characteristics analyzed in this study

| Synoptic conditions | Onset | | | | | | Post-Onset | | | | | | | | | | | | | Recovery | | |
|---|---|---|---|---|---|---|---|---|---|---|---|---|---|---|---|---|---|---|---|---|---|---|
| Months | June 2016 | | | | | | July 2016 | | | | | | | | | | | | | | | |
| Day-D+1 | 20 | 22 | 26 | 27 | 29 | 30 | 01 | 02 | 03 | 04 | 05 | 06 | 07 | 08 | 10 | 11 | 17 | 18 | 19 | 27 | 28 | 29 |
| N° IOP | -- | -- | 03 | -- | 04 | -- | 05 | -- | 06 | -- | 07 | -- | -- | 08 | -- | 09 | -- | 11 | -- | 14 | -- | -- |
| **LLSC at the end of the stratus phase (section 4)** | | | | | | | | | | | | | | | | | | | | | | |
| CBH | 206 | 370 | 204 | 226 | 249 | 174 | 53 | 70 | 91 | 100 | 277 | 147 | 292 | 253 | 299 | 380 | 306 | 338 | 136 | 260 | 206 | 208 |
| Depth | 813 | 499 | 185 | 404 | 381 | 306 | 607 | 320 | -- | 470 | 502 | 452 | 337 | 407 | -- | -- | 384 | 412 | 313 | 385 | 573 | -- |
| Shear$^+$ | 6.7 | 2.2 | 0.1 | 6.0 | 0.8 | 0.4 | 0.5 | 4.5 | -- | 43.3 | 5.5 | 12.3 | 17.2 | 7.1 | -- | -- | -- | -- | 2.6 | -- | -- | -- |
| $\theta^- - 290$ | 7.2 | 7.5 | 6.7 | 7.5 | 7.3 | 7.2 | 6.9 | 6.6 | -- | 6.3 | 7.7 | 7.2 | 8.1 | 7.8 | -- | -- | -- | -- | 6.3 | -- | -- | -- |
| $\Delta\theta_l$ | 3.5 | 2.6 | 1.4 | 2.2 | 2.3 | 1.7 | 3.7 | 1.9 | -- | 4.7 | 4.2 | 2.7 | 1.9 | 2.8 | -- | -- | -- | -- | 2.4 | -- | -- | -- |
| $q^-$ | 16.7 | 15.7 | 15.9 | 16.8 | 16.4 | 16.8 | 17.0 | 16.8 | -- | 16.8 | 16.3 | 17.0 | 16.7 | 16.8 | -- | -- | -- | -- | 16.2 | -- | -- | -- |
| $\Delta q_t$ | -3.0 | -2.1 | -0.1 | -1.4 | -1.3 | -1.5 | -1.3 | -0.6 | -- | -0.1 | -1.5 | -1.0 | -1.3 | -0.7 | -- | -- | -- | -- | -0.2 | -- | -- | -- |
| RAD | 65.9 | 62.7 | 45.2 | 53.3 | 52.4 | 49.7 | 56.0 | 53.4 | -- | 59.2 | 60.8 | 56.5 | 57.5 | 54.9 | -- | -- | -- | -- | 56.5 | -- | -- | -- |
| ENT | -.02 | -9.7 | -0.2 | -3.9 | -6.1 | -10.3 | -0.5 | 1.2 | -- | -6.4 | -11.6 | -0.4 | -7.0 | -2.1 | -- | -- | -- | -- | -1.0 | -- | -- | -- |
| SUBS | -60.9 | -47.9 | -23.8 | -38.8 | -34.4 | -35.5 | -36.5 | -35.7 | -- | -23.5 | -28.9 | -37.6 | -40.5 | -31.6 | -- | -- | -- | -- | -29.3 | -- | -- | -- |
| **LLSC during the convective phase (section 5)** | | | | | | | | | | | | | | | | | | | | | | |
| Scenarios | DD | DD | DC | C | DD | C | C | C | C | C | DD | C | DC | C | DD | DD | DC | DD | C | DC | DD | DC |
| $T_i$ | 0835 | 0730 | 0715 | 0700 | 0810 | 0705 | 0710 | 0655 | 0720 | 0655 | 0805 | 0640 | 0635 | 0740 | 0705 | 0755 | 0910 | 0730 | 0645 | 0745 | 0635 | 0805 |
| $T_b$ | 1105 | 0835 | 1205 | 1135 | 1205 | 1225 | 1310 | 1140 | 1555 | 1105 | 1215 | 1155 | 1255 | 1200 | 1035 | 1010 | 0800 | 0825 | 1220 | 1205 | 0725 | 1235 |

Table A-1: Summary of the LLSC features at the end of the stratus phase (section 4) and during the convective phase (section 5) for the twenty-two occurrences at the Savè supersite analyzed in this study. The Day-D+1 of the night-to-day transition and the eventual corresponding IOP number are indicated. The main synoptic conditions defined by Knippertz et al. (2017), in which they fall are mentioned at the top. The Cloud base height (CBH in m a.g.l) and depth (m) are estimated from the ceilometer and cloud radar measurements. The contribution of wind shear in turbulence production at the cloud top (Shear$^+$, in $10^{-5}$ s$^{-2}$), the thermodynamical properties of the LLSC layer, $\theta^-$ and $\Delta\theta_l$ in K, $q^-$, $\Delta q_t$ and g kg$^{-1}$ as well as the LWP budget terms radiative (RAD), entrainment (ENT) and subsidence (SUBS), in g m$^{-2}$ h$^{-1}$, are derived from the 05:00 UTC standard radiosoundings. They are estimated only for the fourteen cases for which the radiosonde flew into the LLSC layer. The scenario of evolution after sunrise and its characteristic times, the surface-convection influence ($T_i$) and breakup ($T_b$) times are indicated in the format HHMM UTC. C, DC and DD stand for "coupled", "decoupled-coupled" and "decoupled-decoupled" scenarios, respectively. The local time at Savè in (Benin) is UTC + 1 hour.

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
