# Peer review of "Breakup of nocturnal low-level stratiform clouds during the southern West African monsoon season"

_Atmospheric Chemistry and Physics, 2020_

## Referee Comment (RC1) · Anonymous Referee #1 · 12 Aug 2020

Review of "Breakup of nocturnal low-level stratiform clouds during southern West African Monsoon Season"

Authors: Zouzoua et al.

General Comments

This is a very interesting study discussing nocturnal stratiform cloud breakup and mechanisms describing this breakup. A large suite of observations documents several case studies of nocturnal low-level stratiform cloud breakup. This study also evaluates relevant observationally derived quantities describing LWP tendency against numerical simulation results. Finally, this paper describes the nature of coupling (decoupling) for

all available case studies. I learned quite a bit reading this work, noting especially that the intricate detail and analysis are the key strengths of this paper. The results in Section 5 are especially clear and perhaps the biggest strength of this paper. The scientific merit of the work alone merits publication. I do have some concerns, however, with the writing style of this manuscript and found myself at times overwhelmed by too many acronyms, abbreviations and equations in the text, thus needing to refer to multiple copies of this manuscript in order to track down and trace relevant details to pertinent results. I think this is partly because the authors have taken on an extremely difficult task by delving into the many aspects of stratiform cloud breakup. I recommend that the authors take greater care in re-organizing and re-writing portions of the text for clarity (specific comments are below), though I will leave it to their discretion since (again) the scientific quality of this work is excellent. Finally, I believe the authors do a great job of analyzing the available data and frame their results in a very appropriate context.

I refer to specific pages as "P" and lines as "L". For example, "P10, L11" refers to page 10, line 11.

Specific Comments

General comment: I got confused at times, even after reading this twice, keeping track of the large number of acronyms made throughout this text. I see and acknowledge their importance for keeping the paper at an appropriate length, however, I think the authors should take care to re-state some acronyms through the text to clarify what is being discussed.

Section 1: Since this paper describes in great detail many processes responsible for nocturnal cloud maintenance and subsequent breakup, this section (and paper in general) would benefit greatly with some discussion about the land-surface types of the 3 supersites. The a priori knowledge of the typical land surface over this part of the continent may be unknown to several readers, and is especially worth noting since boundary layer heights depend somewhat on the land-surface.

P2, First Paragraph: In this section, you state "However, the diurnal cycle of those clouds is still poorly represented in numerical models" and cite Hannak et al. (2017). This is definitely a strong motivation, but I do not think this point is expanded upon enough in this paragraph. Furthermore, I had some trouble reading through this paragraph as this text seemed disjointed and unclear as to the main motivation. I recommend re-writing this paragraph focusing on the importance of stratiform cloud cover in a global context (e.g. earth's radiation budget, difficulty representing these clouds in climate models; I included a reference that may be of interest and relevant here) and expand upon the processes that make this difficult. Move Fig. 1, the discussion of Fig. 1, and the discussion about "scarce weather monitoring over West Africa" to elsewhere in the text.

Section 2 (beginning on P4): This is overall an excellent review of the relevant processes examined in this paper.

P3, L9: I recommend adding a short description of what a "supersite" is.

P4, L23: "... due to the cooling..." at what level of the atmosphere does this cooling occur? Also, change "their formation" to "cloud formation".

Section 4: I really liked this section and found the intricate level of analysis excellent, though I have to admit – again – I needed to read this multiple times to understand it due mostly to the authors' writing style.

Section 4: I will leave it up to the authors to proceed with this next comment as they see fit. Have you looked into the role of nocturnal cloud thickness as a possible reason why coupling sometimes does (or does not) occur (e.g. Fig. 5)? This is an interesting hypothesis that can (I think) be easily tested using your data. I would expect thicker cloud cover to inhibit surface warming enough to delay or possibly prohibit coupling if other meteorological factors cannot enable the transition. Likewise, could entrainment or precipitation – two sink terms for nocturnal cloud fraction under most conditions – correlate to a delayed coupling? These are questions bred from pure scientific curiosity

based on the results you have shared.

P4, L20: This is an unusual title for a section in a manuscript. Did you mean "State of Art"? Maybe call this section "Review"?

P5, paragraph beginning at L19: There are several recent studies from the Cloud System Evolution over the Trades (CSET) experiment that, I believe, can really strengthen this paragraph and provide additional interesting results to compare & contrast your own results with. I believe intertwining principle results from these works will make your paper more interesting and accessible to research groups studying stratiform cloud breakup elsewhere across the globe, especially since the topic of stratocumulus-to-cumulus (or stratiform cloud breakup) has received increasing attention over the past several years.

Overview of CSET: Mohrmann, J., and Coauthors, 2019: Lagrangian Evolution of the Northeast Pacific Marine Boundary Layer Structure and Cloud during CSET. Mon. Wea. Rev., 147, 4681–4700, https://doi.org/10.1175/MWR-D-19-0053.1.

Lagrangian case studies during CSET: Sarkar, M., P. Zuidema, B. Albrecht, V. Ghate, J. Jensen, J. Mohrmann, and R. Wood, 2020: Observations Pertaining to Precipitation within the Northeast Pacific Stratocumulus-to-Cumulus Transition. Mon. Wea. Rev., 148, 1251–1273, https://doi.org/10.1175/MWR-D-19-0235.1.

Observational perspective of stratiform cloud breakup: Schwartz, M. C., and Coauthors, 2019: Merged Cloud and Precipitation Dataset from the HIAPER GV for the Cloud System Evolution in the Trades (CSET) Campaign. J. Atmos. Oceanic Technol., 36, 921–940, https://doi.org/10.1175/JTECH-D-18-0111.1.

Many of the references cited within this paper may also be relevant and of interest. Finally, this work is related but more peripheral to the main context of your paper, but I think it might be worth reviewing the following study by Schneider et al. which discusses how stratocumulus cloud breakup (over the subtropics) might affect future

climate: Schneider, T., Kaul, C.M. & Pressel, K.G. Possible climate transitions from breakup of stratocumulus decks under greenhouse warming.Nat. Geosci. 12, 163–167 (2019). https://doi.org/10.1038/s41561-019-0310-1. It is interesting to think about how the LLSC breakup paradigm over SWA might change in future climate.

End of P5: Again, this is an overall well-written section. This section seems to come to an abrupt end, however, with no suggestions or links as to how the described relevant dynamical processes relate to the observation studies presented in the remainder of the work.

Section 3.1 Header: I recommend renaming this section as "Instrumentation" instead of "Observational Data Used"

P7, L2: Are missing CTH data from the ceilometer the result of attenuation from optically thick daytime cumulus cloud, or were there frequent instrument malfunctions? This would be useful to know.

Section 3.1: What measurements did the radiosondes collect? And what versions/types of radiosondes were used? This section in general is also lacking descriptions of measurement uncertainties for each instrument. For example, how accurate are the cloud base and cloud top height estimates from the ceilometer? What uncertainty is expected with radiosonde temperature and humidity measurements? I noted some statements of measurement uncertainty and accuracy elsewhere in the text, but these need to be stated here. Finally, presuming meteorological conditions are estimated from the radiosondes, I would put paragraph 2 after the current 3rd paragraph since its unclear at that point in the paper how the authors estimate SHF, LHF, etc.

P11, L11: "Therefore, it has a spatio-temporal variability" this is true but is out of place at this point in the text.

P20, L7: What do you mean by "help us to depart the cases"? Do you mean "differentiate" instead of "depart"? This is confusing and needs clarified since this is obviously

a key science question motivating subsection 4.2.

P20, L12: "Indeed, the crossing of the cloud wets the probe" this sounds very flowery. I recommend rewriting this entire sentence. Suggestion: "Liquid water buildup on the radiosonde's sensors possibly renders some measurements suspect, especially near cloud top."

P20, L23: Again, it is critical to know what the instrument uncertainties (or accuracy) are, such that these over/underestimations have context. This will elucidate the magnitude and seriousness of liquid water condensation on the sensors and subsequent computations using these measurements.

P28, L18-19: "... for which the hydrometeors radar reflectivity from the cloud radar reveals light precipitations above the LLSC layer" The way this sentence is written implies that precipitation is occurring above the cloud layer, which is physically not possible. Did you mean to say that there is precipitation occurring inside the cloud layer? I have a stylistic comment here too: its fine to simply say "collocated cloud radar data revealed precipitation inside the LLSC layer" or something to that effect. "hydrometeors radar reflectivity" is confusing and does not make much sense.

P29, L17: "30% lower" what exactly is 30% lower? the cloud base height? Also, the beginning of this sentence should be "The latter..."

P31, L26: "This could favour the convection in the cloud..." just state "This favors convection which..."

P34, L11: "more significantly impact" is this because the coupled cases generally result in longer lasting cloud cover and therefore decrease the total amount of solar insolation received at the surface? I would be much more specific here since and this statement as written is pretty bold yet a bit hand-wavy.

Figure captions (general comment): It would be helpful to the reader to re-state or spell out acronyms. I found it tough at times to try to dig variable abbreviations from the text

while also trying to follow and learn from the figures.

Technical (Minor) Comments

Title: change "during southern West African" to "during the southern West African"

P1, L17: Change "Savè supersite, in Benin" to "Savè, Benin"

P4, L31: "the maximum of wind speed" → "the maximum wind speed" P2, L13-15: The two sentences here, beginning with "Figure 1 gives...", are interesting details but, in my opinion, would be more effective if discussed in Section 2.

P3, L11: "at Savè supersite" → "at the Savè supersite".

P5, L21: Eliminate the work "essentially".

P6, L17: "because of a good data coverage" → "because of good data coverage"

P6, L13: Recommend rewriting this sentence to "Data acquired at the Savè supersite offer nearly continuous information on atmospheric conditions." Sky coverage (clear or cloudy skies) is implied when you mention atmospheric conditions.

P7, L3: comma use, remove the second commas after each "and".

P7, L8: "the radiosoundings" → radiosondes.

P7, L14: "data were smoothed by averaging with final vertical resolution..." → "data were averaged to a final vertical resolution..."

P9, L19-20: Recommended rewording "...of horizontally-homogeneous stratocumulus cloud cover maintained by vertical mixing, which is driven by convective turbulence and cloud-top radiative cooling (references)". The current version of this sentence reads awkwardly.

P11, L18: Change this sentence to "For simplicity and due to a lack of precise estimate, we assume..."

P11, L28: "In sum, ..." → "In summary, ..."

P15, L18: "This supposes that, the potential..." → "This evidence suggests potential early morning coupling..."

P27, L4: "...presented in Fig. 8, and summarizing the..." → "...presented in Fig. 8 which summarize...". Also please remove the word "Eventually" from the beginning of this sentence.

P27, L6: "... is larger than 95% whatever the case, and..." I am extremely confused by "whatever the case" means here.

P27, L13: remove the word "firstly"

P28, L22: "here after" → "hereafter"

---

## Referee Comment (RC2) · Anonymous Referee #2 · 17 Aug 2020

This study attempts to understand the evolutions of daytime low-lying stratiform clouds during southern West Africa monsoon seasons. The analyses are primarily based on ground-based observations of about two dozens cases collected during the DACCIWA field campaign. Although the size of the samples is not large enough to conduct statistically robust analyses, the authors did an overall good job of taking full advantage of the cases by conducting systematic and all-round analyses. In particular, the budget analyses of liquid water path (LWP) are conducted, which has been considered very challenging for observational studies since many budget terms are difficult to quantify with observations. In that regard, the authors' patience to quantify each LWP budget terms and their potential uncertainties is very impressive (although uncertainties are

still considerable). Overall, I think it is a high-quality manuscript with well-organized presentations, mostly solid analyses, and useful conclusions. This makes me believe that the manuscript should be eventually suitable for being published in ACP. However, there are several major issues that must be addressed before I recommend acceptance. I detail them below:

(1) Insufficient treatment of radiative cooling term (RAD) quantification RAD is the dominant term controlling the convective overturning before the early morning, as also recognized by the authors. However, the equations (Eq. 2 and 3) used to quantify RAD in this study are too rough. As shown by Zheng et al. (2019), the RAD is most sensitive to two parameters: cloud optical thickness and moisture loading in the free atmosphere. If high clouds are present, the RAD will weaken significantly (e.g. Christensen et al., 2013). Even though the free-tropospheric moisture loading can be somewhat accounted for in Eq. (2) (the IWP), the cloud optical thickness and higher clouds can also modulate the RAD considerably. The blackbody assumption is only always valid for not-too-thick stratiform clouds (Zheng et al., 2019). The authors show that the RAD varies very little ( $\sim$  5 Wm-2), which could be artificial consequence of the two assumptions behind the equations (i.e. blackbody and no high clouds). Thus, given the significant role of RAD, it should be worthwhile to use a radiative transfer model instead. All inputs for the model are available from the observations: cloud-base and -top heights and soundings. Running it is computationally cheap.

(2) Inappropriate classification of the scenario of DD I am very reluctant to consider the clouds in Fig.10 c as "decoupled throughout the day". There are three possibilities for this case: (1) initially decoupled clouds remain decoupled and surface-heating-driven cumulus clouds start to form underneath it. If they don't interact, the upper-layer clouds are decoupled and the bottom clouds are coupled; (2) if they interact, they form the cumulus-coupled stratocumulus-topped boundary layer such as those in down-stream subtropical oceans; (3) If the initially decoupled clouds left, this case is simply regular
continental shallow cumulus that are, by definition, coupled.

All the above-stated cloud regimes are possible. Thus, it is a little bit misleading to call all of them "decoupled throughout". I would suggest either renaming it or adding additional discussions to clarify the definition of the decoupling.

(3) other comments: - Page 5, Line 25: there are earlier literatures form the ASTEX campaign that is the first attempt to study the SC-to-cu transitions. - Figure 2 and other figures: it should be helpful to use local time as well, which makes the readers easier to think of the problem from a diurnal cycle perspective. - Page 10-11: some discussions on what determines the RAD is useful. (check the work by Zheng et al., 2019) - Page 12, Line1: large-scale subsidence is commonly obtained from reanalysis data. Not very accurate, but better than nothing. - Section 4.1 as a whole: this section is centered on the difference between coupling and decoupling, however, what may cause the decoupling/coupling in the first place is not discussed in detail. There are several influential factors: cloud-top cooling itself (Nicholl 1984), precipitation (this is not important in your case), "deepening warming" decoupling (Bretherton and Wyant, 1997), and warm thermal advection (Zheng and Li, 2019). It may be more enlightening to discuss your results in the context of these potential influential controllers. - Page 22, Line 15: again, it could be due to too simple treatment of RAD. - Figure 13: there are too many symbols, making the readers hard to recognize each of them. This defeats the purpose of using a diagram for illustrations. Try to use process-based cartoons (e.g. the one from Wood 2012).

References:

Zheng, Y., Rosenfeld, D., Zhu, Y., & Li, Z. (2019). Satellite-based estimation of cloud top radiative cooling rate for marine stratocumulus. Geophysical Research Letters, 46(8), 4485-4494.

Nicholls, S. (1984), The dynamics of stratocumulus: Aircraft observations and comparisons with a mixed layer model, Quarterly Journal of the Royal Meteorological Society, Interactive comment

110(466), 783-820.

Bretherton, C. S., and M. C. Wyant (1997), Moisture transport, lower-tropospheric stability, and decoupling of cloud-topped boundary layers, Journal of the atmospheric sciences, 54(1), 148-167.

Zheng, Y., & Li, Z. (2019). Episodes of warm-air advection causing cloud-surface decoupling during the MARCUS. Journal of Geophysical Research: Atmospheres, 124(22), 12227-12243.

---

## Author Comment (AC1) · 5 Oct 2020

Dear reviewer 2,

We thank the reviewer for his/her helpful suggestions, which led to significant improvements of our paper. Below we detailed how his/her comments are addressed in the revised version of the paper. The major corrections of the paper are cited here in *italic. We refer to specific pages by "P" and lines by "L". For example, "**P1, L1**" refers to page 1, line 1.*

(1) Insufficient treatment of radiative cooling term (RAD) quantification RAD is the dominant term controlling the convective overturning before the early morning, as also recognized by the authors. However, the equations (Eq. 2 and 3) used to quantify RAD in this study are too rough. As shown by Zheng et al. (2019), the RAD is most sensitive to two parameters: cloud optical thickness and moisture loading in the free atmosphere. If high clouds are present, the RAD will weaken significantly (e.g. Christensen et al., 2013). Even though the free-tropospheric moisture loading can be somewhat accounted for in Eq. (2) (the IWP), the cloud optical thickness and higher clouds can also modulate the RAD considerably. The blackbody assumption is only always valid for not-too-thick stratiform clouds (Zheng et al., 2019). The authors show that the RAD varies very little (_ 5 Wm-2), which could be artificial consequence of the two assumptions behind the equations (i.e. blackbody and no high clouds). Thus, given the significant role of RAD, it should be worthwhile to use a radiative transfer model instead. All inputs for the model are available from the observations: cloud-base and -top heights and soundings. Running it is computationally cheap.

We thank the reviewer for this valuable suggestion to use a radiative transfer code. However, the water or ice content, the base and the summit of each cloud layers is needed in the radiative transfer code in order to take into account the higher clouds effect. This information is missing for the DACCIWA campaign, since only integrated LWP, the LLSC base and top heights are available. So the use of the radiative code does not fully answer the reviewer comment. Despite this, the SBDART (Santa Barbara DISORT Atmospheric Radiative Transfer; Ricchiazzi et al., 1998) model is now used in our study to estimate the radiative cooling over the LLSC layer at the end of the stratus phase, based on radiosonde, ceilometer and cloud-radar measurements. The LLSC optical thickness is determined by a parameterized LWP. The higher clouds impact is partly taken into account through vertical profiles of temperature and relative humidity given by the radiosonde but an emissivity of clear air is applied to these thermodynamical characteristics. This limitation is further discussed in the paper. We obtain higher values (+ 15 W m$^{-2}$ in average) of cloud-top radiative cooling than previously, but the standard deviation among the cases is still of 5 W m$^{-2}$ and no difference can be noticed between coupled and decoupled LLSC.

The text was modified in several places to include the SBDART radiative code description, and the discussion of the results:
**P11-12**: *"The term RAD (Eq. 1.d) is retrieved from the vertical profiles of upwelling and downwelling radiative fluxes which are computed by using the Santa Barbara DISORT Atmospheric Radiative Transfer (SBDART) model (Ricchiazzi et al., 1998). This software tool, which solves the radiative transfer equation for a plane-parallel*

*atmosphere in clear and cloudy conditions, was used in the studies of Babić et al. (2019a) and Adler et al. (2019) to estimate the temperature tendency due to radiative interactions during the LLSC diurnal cycle. For our simulations, the model configuration was very similar to that used in these studies. We prescribed 65 vertical input levels with a vertical resolution of 50 m below 2 km a.g.l, 200 m between 2 and 5 km a.g.l, and, 1 km above 5 km a.g.l. The vertical profiles of air pressure, temperature and water vapour density as well as the integrated water vapour are based on 05:00 UTC standard radiosounding data. The cloud optical thickness, which varies with its water and ice content, is required to describe a cloud layer in the SBDART model. Yet, the LWP provided by the microwave radiometer deployed at Savè supersite (Wieser et al., 2016) includes all the existing cloudy layers, and also is not available for five of our selected cases. Therefore, the LLSC optical thickness is determined from a parameterized LWP (Eq. 2), by assuming an adiabatic cloudy layer in which the liquid water mixing ratio ($q_l$) increases linearly (van der Dussen et al., 2014; Pedruzo-Bagazgoitia et al., 2020). The downwelling longwave radiations from potential mid-level and high-level clouds may reduce the radiative cooling at the stratocumulus top (e.g. Christensen et al., 2013). However, the cloud layers above the LLSC (base, top and water content) cannot be precisely described in the SBDART model from the available data set. Thus, the higher clouds radiative effect is not directly included in our estimate of downwelling radiative fluxes, but it is partially taken into account through vertical profiles of temperature and relative humidity given by the radiosonde. As the shortwave radiations are zero before the sunrise, only the longwave range, 4.5-42 μm with spectral resolution of 0.1μm (Babić et al., 2019a), was selected for radiative fluxes calculations. For all the cases, the vertical optical depth of ABL aerosol is fixed to 0.38, which corresponds to the average value of the measurements performed with a sun photometer in June and July 2016 at Savè."*

(2) Inappropriate classification of the scenario of DD I am very reluctant to consider the clouds in Fig.10 c as "decoupled throughout the day". There are three possibilities for this case: (1) initially decoupled clouds remain decoupled and surface-heating driven cumulus clouds start to form underneath it. If they don't interact, the upper-layer clouds are decoupled and the bottom clouds are coupled; (2) if they interact, they form the cumulus-coupled stratocumulus-topped boundary layer such as those in downstream subtropical oceans; (3) If the initially decoupled clouds dissipate rapidly after decoupling, with only the underlying cumulus clouds left, this case is simply regular continental shallow cumulus that are, by definition, coupled.

All the above-stated cloud regimes are possible. Thus, it is a little bit misleading to call all of them "decoupled throughout". I would suggest either renaming it or adding additional discussions to clarify the definition of the decoupling.

We thank the reviewer for this comment. We fully agree that the three possibilities for scenario DD may occur. However, as stated in the paper, the scenario description is based on temporal changes of surface-based LCL and cloud base height measured by the ceilometer. From this point of view, in the scenario DD, the LLSC remains decoupled from the surface and thermally-driven (and coupled) shallow cumulus forms below it at the beginning of the convective phase. We are not able to test if the top of this underlying shallow cumulus interacts or not with the LLSC. So we kept the same name (DD) for this case. However, we completed the discussion about it.

The previous sentence *"In such conditions, the underlying cumulus clouds act to intermittently and locally couple the stratocumulus layer with the surface (Wood, 2012)."* was replaced by a more complete comment as suggested by the reviewer, **P29, L24** : *"In the case where the two cloud layers are superimposed, two possibilities may occur: (i) the underlying surface-convection driven cumulus cloud do not interact with the LLSC which remains decoupled from the surface, (ii) the underlying cumulus clouds develop vertically, reach the LLSC layer, and act to intermittently and locally couple it with the surface (Wood, 2012)."*

We moderated the statement in several sentences like this one, **P30, L3**, *"One can wonder what conditions lead the LLSC to either be coupled to the surface in the scenario DC, or remains POSSIBLY decoupled with the formation of an underlying cumulus layer in the scenario DD."*

The previous sentence, in the Abstract, *"In the eight remaining cases, the stratiform cloud remains decoupled from the surface all along its life cycle.",* is now **P2, L1***: "In the eight remaining cases, the stratiform cloud remains HYPOTHETICALLY decoupled from the surface all along its life cycle, since the cloud base remains separated from the condensation level."*

(3) Other comments: - Figure 2 and other figures: it should be helpful to use local time as well, which makes the readers easier to think of the problem from a diurnal cycle perspective.

We thank the reviewer for this suggestion. We indicate in the section 3, **P7-L12**, that the local time at Savè, Benin is UTC +1 hour. In the revised version, this local time is repeated in the caption of Figures 2, 10 and 13.

- Page 10-11: some discussions on what determines the RAD is useful (check the work by Zheng et al., 2019).

We thank the reviewer for this comment. The radiative transfer across the stratocumulus layer is discussed in section 2; the text was modified to make it clear as follow, **P5-L15**: *"During night-time, the longwave radiative cooling at the stratocumulus top is the leading process governing its maintenance. This cooling occurs because the cloud droplets emit more infrared radiation towards the free troposphere than they receive from the drier air above. It is modulated by cloud-top temperature, cloud optical thickness, thermodynamic and cloudy conditions in the free troposphere (Siems et al., 1993; Wood, 2012; Christensen et al., 2013; Zheng et al., 2019)."*

- Page 12, Line1: large-scale subsidence is commonly obtained from reanalysis data. Not very accurate, but better than nothing.

We agree with the reviewer and actually tried to use reanalysis data from the beginning. As mentioned in Pedruzo-Bagazgoitia et al. (2020), the large scale vertical velocity from reanalysis products present strong temporal and vertical variability, especially on early morning hours. We observed the same behaviour when we tried to use the ERA5 reanalysis products. Beside this, we observed a steady LLSC top at the end of the stratus phase in many cases. Consequently, we decided to use the Lilly (1968) assumption that implies the same order of magnitude between parameterized entrainment and subsidence velocities at the LLSC top.

The text is now, **_P12, L25_**: *"For the term SUBS (Eq. 1.e), we have no possibility of estimating precisely the large scale subsidence at the LLSC top. One possibility is to consider evaluations from models or re-analyses. However, we decided to discard this approach, because the subsidence profiles from regional simulations with Consortium for Small-Scale Modelling (COSMO) or from ERA-interim and ERA-5 reanalyses showed a very high temporal variability and a strong lack of coherence among the different cases. According to the cloud-radar CTH estimates, the LLSC top is often stationary at the end of the stratus phases during DACCIWA. This feature has been observed (Adler et al., 2019; Babić et al., 2019a; Dione et al., 2019) but also simulated by Pedruzo-Bagazgoitia et al. (2020). Based on the LLSC top stationarity at the time of our LWP budget analysis, $w_{s,CTH}$ is estimated following Lilly (1968):"*

$$\frac{\partial CTH}{\partial t} = w_{s,CTH} + w_e \approx 0 \qquad\qquad (6)$$

*"*

- Section 4.1 as a whole: this section is centered on the difference between coupling and decoupling, however, what may cause the decoupling/coupling in the first place is not discussed in detail. There are several influential factors: cloud-top cooling itself (Nicholl 1984), precipitation (this is not important in your case), "deepening warming" decoupling (Bretherton and Wyant, 1997), and warm thermal advection (Zheng and Li, 2019). It may be more enlightening to discuss your results in the context of these potential influential controllers.

We thank the reviewer for this suggestion, and we hope to have improved the text. The section 4 has been deeply modified; a section 4.3 has been added, **_P26-27_**, to discuss the results presented in section 4.1 and 4.2 about the relevant processes which are able to couple the LLSC during the stratus phase. In summary, none of these processes was clearly pointed out as responsible for the coupling during this phase and a combination of several of them, each with a small effect, should be considered.

- Page 22, Line 15: again, it could be due to too simple treatment of RAD.

We do agree with the reviewer. The use of SBDART certainly gives a better treatment of RAD but still not complete, since the higher clouds are not fully taken into account. This is discussed in the revised version **_P24, L3_**:

*"We find only a 5 W m$^{-2}$ standard deviation for the radiative cooling at the LLSC top and no particular difference between cases C and D. This very low standard deviation may be due to the conditions which remained very steady from one case to the other, but may also be underestimated because the higher clouds impact is not fully included in the radiative fluxes estimate. In order to evaluate the error due to the temperature underestimation above the LLSC top, SBDART is run with the measured and a corrected temperature profile, while the other inputs remain unchanged. The correction of the potential temperature vertical profile consists in a linear tendency between the measured θ plus a 1.2K correction right above the CTH, and the measured θ at 800 m, where we consider that the radiosonde sensor is no more affected by the cloud crossing. The cloud-top radiative cooling estimated by SBDART with this corrected temperature vertical profile is larger by less than 2 W m$^{-2}$."*

- Figure 13: there are too many symbols, making the readers hard to recognize each of them. This defeats the purpose of using a diagram for illustrations. Try to use process-based cartoons (e.g. the one from Wood 2012).

We thank the reviewer for this valuable suggestion. Process-based cartoons are now used in Figure 13 to illustrate the different scenarios, **_P37_**.

Adler, B., Babić, K., Kalthoff, N., Lohou, F., Lothon, M., Dione, C., Pedruzo-Bagazgoitia, X. and Andersen, H.: Nocturnal low-level clouds in the atmospheric boundary layer over southern West Africa: an observation-based analysis of conditions and processes, Atmospheric Chemistry and Physics, 19(1), 663–681, doi:10.5194/acp-19-663-2019, 2019.

Babić, K., Adler, B., Kalthoff, N., Andersen, H., Dione, C., Lohou, F., Lothon, M. and Pedruzo-Bagazgoitia, X.: The observed diurnal cycle of low-level stratus clouds over southern West Africa: a case study, Atmospheric Chemistry and Physics, 19(2), 1281–1299, doi:10.5194/acp-19-1281-2019, 2019.

Christensen, M. W., Carrió, G. G., Stephens, G. L. and Cotton, W. R.: Radiative Impacts of Free-Tropospheric Clouds on the Properties of Marine Stratocumulus, Journal of the Atmospheric Sciences, 70(10), 3102–3118, doi:10.1175/JAS-D-12-0287.1, 2013.

Dione, C., Lohou, F., Lothon, M., Adler, B., Babić, K., Kalthoff, N., Pedruzo-Bagazgoitia, X., Bezombes, Y. and Gabella, O.: Low-level stratiform clouds and dynamical features observed within the southern West African monsoon, Atmos. Chem. Phys., 19(13), 8979–8997, doi:10.5194/acp-19-8979-2019, 2019.

van der Dussen, J. J., de Roode, S. R. and Siebesma, A. P.: Factors Controlling Rapid Stratocumulus Cloud Thinning, J. Atmos. Sci., 71(2), 655–664, doi:10.1175/JAS-D-13-0114.1, 2014.

Lilly, D. K.: Models of cloud-topped mixed layers under a strong inversion, Q.J.R. Meteorol. Soc., 94(401), 292–309, doi:10.1002/qj.49709440106, 1968.

Pedruzo-Bagazgoitia, X., de Roode, S. R., Adler, B., Babić, K., Dione, C., Kalthoff, N., Lohou, F., Lothon, M. and Vilà-Guerau de Arellano, J.: The diurnal stratocumulus-to-cumulus transition over land in southern West Africa, Atmos. Chem. Phys., 20(5), 2735–2754, doi:10.5194/acp-20-2735-2020, 2020.

Ricchiazzi, P., Yang, S., Gautier, C. and Sowle, D.: SBDART: A Research and Teaching Software Tool for Plane-Parallel Radiative Transfer in the Earth's Atmosphere, Bull. Amer. Meteor. Soc., 79(10), 2101–2114, doi:10.1175/1520-0477(1998)079<2101:SARATS>2.0.CO;2, 1998.

Siems, S. T., Lenschow, D. H. and Bretherton, C. S.: A Numerical Study of the Interaction between Stratocumulus and the Air Overlying It, J. Atmos. Sci., 50(21), 3663–3676, doi:10.1175/1520-0469(1993)050<3663:ANSOTI>2.0.CO;2, 1993.

Wieser, A., Adler, B. and Deny, B.: DACCIWA field campaign, Savè super-site, Thermodynamic data sets, , doi:10.6096/dacciwa.1659, 2016.

Wood, R.: Stratocumulus Clouds, Mon. Wea. Rev., 140(8), 2373–2423, doi:10.1175/MWR-D-11-00121.1, 2012.

Zheng, Y., Rosenfeld, D., Zhu, Y. and Li, Z.: Satellite- Based Estimation of Cloud Top Radiative Cooling Rate for Marine Stratocumulus, Geophys. Res. Lett., 46(8), 4485–4494, doi:10.1029/2019GL082094, 2019.

---

## Author Comment (AC2) · 5 Oct 2020

General comment: I got confused at times, even after reading this twice, keeping track of the large number of acronyms made throughout this text. I see and acknowledge their importance for keeping the paper at an appropriate length, however, I think the authors should take care to re-state some acronyms through the text to clarify what is being discussed.

We fully understand this difficulty and we tried to re-state the different acronyms through the text and figure captions.

Section 1: Since this paper describes in great detail many processes responsible for nocturnal cloud maintenance and subsequent breakup, this section (and paper in general) would benefit greatly with some discussion about the land-surface types of the 3 supersites. The a priori knowledge of the typical land surface over this part of the continent may be unknown to several readers, and is especially worth noting since boundary layer heights depend somewhat on the land-surface.

We thank the reviewer for this remark. We added the climatic zones of West Africa affected by the LLSC in the introduction P2, L12-15: "During the West Africa monsoon season, the LLSC form frequently at night over a region extending from Guinean coast to several hundred kilometres inland (van der Linden et al., 2015), which includes the coastal, Sudanian and Sudanian-Sahelian climatic zones (Emetere, 2016)."

In addition, this statement in section 3: "The ground sites were located at roughly the same distance from the Guinean coast (200 km in land) but with different topography (Kalthoff et al., 2018)", has been modified as follow, P6, L25: "The DACCIWA supersites were located at roughly the same distance from the Guinean coast (200 km in land, Fig. 1), between the coastal and the Sudanian areas, but with a different topography (Kalthoff et al., 2018). The supersites are part of the savannah ecosystem, where grassland is intercut with crops and degraded forest."

P2, First Paragraph: In this section, you state "However, the diurnal cycle of those clouds is still poorly represented in numerical models" and cite Hannak et al. (2017). This is definitely a strong motivation, but I do not think this point is expanded upon enough in this paragraph. Furthermore, I had some trouble reading through this paragraph as this text seemed is jointed and unclear as to the main motivation. I recommend re-writing this paragraph focusing on the

importance of stratiform cloud cover in a global context (e.g. earth's radiation budget, difficulty representing these clouds in climate models; I included a reference that may be of interest and relevant here) and expand upon the processes that make this difficult. Move Fig. 1, the discussion of Fig. 1, and the discussion about "scarce weather monitoring over West Africa" to elsewhere in the text.

We thank the reviewer for this comment. The paragraph was modified:

- 1/ The comment on figure 1 was moved in the next paragraph.
- 2/ We improved the first paragraph of section 1, as follow, P2:

"The low-level stratiform clouds (LLSC) are Earth's most common cloud type (Wood, 2012). During the West Africa monsoon season (WAM), the LLSC form frequently at night over a region extending from Guinean coast to several hundred kilometres inland (van der Linden et al., 2015), which includes the coastal, Sudanian and Sudanian-Sahelian climatic zones (Emetere, 2016). The LLSC coverage persists for many hours during the following day, reducing the incoming solar radiation, impacting the surface energy budget and related processes such as the diurnal cycle of the atmospheric boundary layer (ABL) (Schuster et al., 2013; Adler et al., 2017; Knippertz et al., 2017). However, the diurnal cycle of those clouds is still poorly represented in numerical weather and climate models, especially over West Africa (Hannak et al., 2017). Indeed, their lifetime is generally underestimated in the numerical simulations, causing high incoming solar radiation at the surface in this region where the meteorological conditions are governed by convection activities and by surface thermal and moisture gradients (Knippertz et al., 2011). That could be an important factor for which the forecasts of WAM features still have a poor skill (Hannak et al., 2017). Therefore, a better understanding of the processes behind LLSC over SWA is useful to improve the numerical weather prediction and climate projection quality. Due to the scarce weather monitoring network over West Africa, the first studies addressing the LLSC over this region were mostly conducted with satellite images and traditional synoptic observations (Schrage and Fink, 2012; van der Linden et al., 2015), as well as with numerical simulations at regional scale (Schuster et al., 2013; Adler et al., 2017; Deetz et al., 2018). They emphasized that the physical processes, spanning from local to synoptic scale such as, horizontal advection of cold air associated to WAM, lifting induced by topography, gravity waves or shear-driven turbulence, are relevant for the LLSC formation during the night. However, the LLSC evolution after the sunrise received little attention."

P3, L9: I recommend adding a short description of what a "supersite" is.

The sentence has been modified to define a supersite as a site gathering a comprehensive set of instrumentation, P3, L8-10: "To this end, three so-called "supersites", which gather a large set of complementary instruments, were installed at Kumasi (6.68° N, 1.56° E) in Ghana, Savè (8.00° N, 2.40° W) in Benin, and Ile-Ife (7.55° N, 4.56° W) in Nigeria (Fig. 1)."

P4, L23: "... due to the cooling..." at what level of the atmosphere does this cooling occur? Also, change "their formation" to "cloud formation".

The sentence has been corrected and completed as follow P4, L26-28: "The increase of relative humidity (Rh) within the ABL leading to saturation and LLSC formation is due to the cooling which mainly occurs during the stable and the jet phases in the monsoon layer, up to around 1.5 km above ground level (a.g.l.)."

Section 4: I really liked this section and found the intricate level of analysis excellent, though I have to admit - again - I needed to read this multiple times to understand it due mostly to the authors' writing style.

We thank the reviewer for this comment. The section 4 was deeply modified and, we hope, improved. We added a section  $(4.3, \underline{P26})$  in order to discuss the different processes possibly responsible for the LLSC coupling with the surface during the stratus phase.

Section 4: I will leave it up to the authors to proceed with this next comment as they see fit. Have you looked into the role of nocturnal cloud thickness as a possible reason why coupling sometimes does (or does not) occur (e.g. Fig. 5)? This is an interesting hypothesis that can (I think) be easily tested using your data. I would expect thicker cloud cover to inhibit surface warming enough to delay or possibly prohibit coupling if other meteorological factors cannot enable the transition. Likewise, could entrainment or precipitation – two sink terms for nocturnal cloud fraction under most conditions – correlate to a delayed coupling? These are questions bred from pure scientific curiosity based on the results you have shared.

We had the same questions as the reviewer and all the reviewer suggestions were tested. We know that it is a bit frustrating but no clear reason explaining the cloud coupling during the stratus phase was highlighted and so only hypotheses were suggested. Concerning the cloud thickness, we showed in Figure 6 that there are no obvious differences between coupled and decoupled LLSC thickness. We were not able to compare the liquid water path of coupled and decoupled LLSC, which could also play an important role.

However it is not a question of convection at that time of the day, since section 4 shows that the stratus phase ends more or less when the convection starts.

The entrainment at the end of the stratus phase is small and very similar in coupled and decoupled cases, but we were not able to check if it was also the case before the coupling. The estimation of the entrainment term along the stratus phase was not possible either.

At last, the precipitation hypothesis could be excluded since only LLSC without precipitation recorded at surface are considered. Of course, precipitation above the LLSC from higher clouds could not be investigated but is one of the hypotheses.

P4, L20: This is an unusual title for a section in a manuscript. Did you mean "State of Art"? Maybe call this section "Review"?

We actually meant "State of Art". "*Review*" is now the title.

P5, paragraph beginning at L19: There are several recent studies from the Cloud System Evolution over the Trades (CSET) experiment that, I believe, can really strengthen this

paragraph and provide additional interesting results to compare & contrast your own results with. I believe intertwining principle results from these works will make your paper more interesting and accessible to research groups studying stratiform cloud breakup elsewhere across the globe, especially since the topic of stratocumulus-to-cumulus (or stratiform cloud breakup) has received increasing attention over the past several years.

We thank the reviewer for these recent studies based on CSET field experiment. They are now cited as many others previous studies addressing the stratocumulus-tocumulus transition in marine conditions. These studies focused on aerosol microphysical role in the scenario of transition from stratocumulus-to-cumulus. Assessing the impact of low-troposphere aerosol loading on the LLSC diurnal cycle is not among the objectives of our study. But, this aspect will be addressed in future research work based on DACCIWA dataset. Thus, this perspective was added in section 6, P36, L25: "The aerosol loading in the low-troposphere is a potential factor controlling the LLSC evolution and lifetime (Deetz et al., 2018; Mohrmann et al., 2019). The airborne measurements of low-cloud properties over SWA during DACCIWA (Flamant et al., 2017) could be used to assess the microphysical role for aerosol in the LLSC evolution scenario. This may help to differentiate the scenarios DC and DD."

End of P5: Again, this is an overall well-written section. This section seems to come to an abrupt end, however, with no suggestions or links as to how the described relevant dynamical processes relate to the observation studies presented in the remainder of the work.

We thank the reviewer for this comment. A sentence was added at the end of the paragraph to better link the LES study with the present observational work. P6, L7: "Since the LES made by Pedruzo-Bagazgoitia et al. (2020) are set with atmospheric and surface conditions measured at Savè during the DACCIWA campaign, some simplifying assumptions used in our study are based on their results, and the simulated and observational results are compared."

Section 3.1 Header: I recommend renaming this section as "Instrumentation" instead of "Observational Data Used"

We thank the reviewer for this suggestion. The modification has been done; "*Instrumentation*" is now the title.

P7, L2: Are missing CTH data from the ceilometer the result of attenuation from optically thick daytime cumulus cloud, or were there frequent instrument malfunctions? This would be useful to know.

Section 3.1: What measurements did the radiosondes collect? And what versions/ types of radiosondes were used? This section in general is also lacking descriptions of measurement uncertainties for each instrument. For example, how accurate are the cloud base and cloud top height estimates from the ceilometer? What uncertainty is expected with radiosonde temperature and humidity measurements? I noted some statements of measurement uncertainty and accuracy elsewhere in the text, but these need to be stated here. Finally,

presuming meteorological conditions are estimated from the radiosondes, I would put paragraph 2 after the current 3rd paragraph since its unclear at that point in the paper how the authors estimate SHF, LHF, etc.

We agree with the reviewer that some indications were missing in this section. The paragraph has been deeply modified and includes now:

1/ The reason why some CTHs are missing, P7, L24: "Unfortunately, several values of CTHs are missing, particularly during daytime for many selected cases, due to the retrieval technique limitation."

2/ Radiosondes sensors measurements accuracy, P7, L26: "The thermodynamical and dynamical characteristics of the low troposphere are retrieved from the radiosondes of the MODEM radiosounding system. The MODEM radiosonde collects, every second (which corresponds to a vertical resolution of 4-5 m), the air temperature and relative humidity, and the probe GPS localization from which horizontal wind speed components, altitude and pressure are deduced (Derrien et al., 2016). The sensors accuracy is 0.2 °C, 2 % and 0.01 m for temperature, relative humidity and GPS localization respectively."

3/ Information on the data acquired by the surface station, P8, L5: "The meteorological conditions at the surface (temperature, relative humidity and pressure of the air at 2 m a.g.l), and some terms of the surface energy budget (net radiative flux  $(R_{n0})$ , sensible heat (SHF0) and latent heat (LHF0) fluxes at 4 m a.g.l) were continuously acquired. SHF0 and LHF0 are deduced from high-frequency (20 Hz) measurements processed with Eddy-covariance methods by using the TK3.11 software (Mauder et al., 2013)."

P11, L11: "Therefore, it has a spatio-temporal variability" this is true but is out of place at this point in the text.

We meant to say that despite the spatial and temporal variability of A, this parameter is very often considered as a constant. The sentences were modified, P12, L8: "A varies with  $\Delta \theta_1$ ,  $\Delta q_t$ , wind shear at the cloud top, surface turbulent fluxes and cloud microphysical processes via the buoyancy flux vertical profile (Stevens et al., 2005; Stevens, 2006). Despite the spatial and temporal variability of A, its value is generally fixed and treated as a constant parameter in several research studies (e.g. van Zanten et al., 1999; van der Dussen et al., 2014)."

P20, L7: What do you mean by "help us to depart the cases"? Do you mean "differentiate" instead of "depart"? This is confusing and needs clarified since this is obviously a key science question motivating subsection 4.2.

We apologize for this word which was misleading. The sentence was modified as follow, **P21**, **L7**: "Does the LWP budget analysis help us to differentiate the cases C and D?"

P20, L12: "Indeed, the crossing of the cloud wets the probe" this sounds very flowery. I recommend rewriting this entire sentence. Suggestion: "Liquid water buildup on the radiosonde's sensors possibly renders some measurements suspect, especially near cloud top."

We thank the reviewer for this suggestion. *The correction was made accordingly*, P21, L13.

P20, L23: Again, it is critical to know what the instrument uncertainties (or accuracy) are, such that these over/underestimations have context. This will elucidate the magnitude and seriousness of liquid water condensation on the sensors and subsequent computations using these measurements.

The accuracy of the radiosonde sensors is now introduced in section 3. See response to previous comment.

P28, L18-19: "... for which the hydrometeors radar reflectivity from the cloud radar reveals light precipitations above the LLSC layer" The way this sentence is written implies that precipitation is occurring above the cloud layer, which is physically not possible. Did you mean to say that there is precipitation occurring inside the cloud layer? I have a stylistic comment here too: its fine to simply say "collocated cloud radar data revealed precipitation inside the LLSC layer" or something to that effect. "hydrometeors radar reflectivity" is confusing and does not make much sense.

We thank the reviewer for this suggestion. The paragraph is certainly unclear. There are sometimes higher clouds above the LLSC. In that case, the radar reveals light precipitation between the higher clouds and the LLSC which was not recorded at surface. The sentence was modified, P31, L23:

"The latest breakup time occurring at 16:00 UTC corresponds to the 02-03 July 2016 case for which the collocated radar reveals light precipitations from higher clouds, above the LLSC layer, during the first hours of the convective phase (not shown) while nothing was recorded by the surface rain gauge."

P29, L17: "30% lower" what exactly is 30% lower? the cloud base height? Also, the beginning of this sentence should be "The latter..."

We thank the reviewer for this comment. The sentence was clarified, P32, L21: "The LLSC breakup time impacts the radiative budget at surface over the day, then the surface fluxes, and consequently, the vertical development of the ABL, as shown by Lohou et al., 2020. They estimated that the ABL height is about 900 m when the LLSC

breaks up at 09:00 UTC and is 30% lower when the LLSC breaks up at 12:00 UTC. Consequently, one can expect a quite different vertical development of the ABL in C/DC cases than in DD cases."

P31, L26: "This could favour the convection in the cloud..." just state "This favours convection which..."

We thank the reviewer for this suggestion. The sentence was corrected, P34, L25: "This favours convection in the LLSC which enhances the entrainment, at the expense of the cloud moistening by the underlying turbulent mixing."

P34, L11: "more significantly impact" is this because the coupled cases generally result in longer lasting cloud cover and therefore decrease the total amount of solar insolation received at the surface? I would be much more specific here since and this statement as written is pretty bold yet a bit hand-wavy.

We fully agree with this comment. The discussion concerning the LLSC impact on surface energy budget is now, P36, L15: "It determines the LLSC lifetime and the way by which the transition towards shallow convective clouds occurs. The coupled LLSC last longer (breakup time at 12:00 in average) than decoupled cases (breakup time at 10:00 UTC in average). According to Lohou et al. (2020), such a difference in breakup time leads to a reduction of about 15% of net radiation at surface and of ABL vertical development during the day, for coupled cases compared to decoupled one."

Figure captions (general comment): It would be helpful to the reader to re-state or spell out acronyms. I found it tough at times to try to dig variable abbreviations from the text while also trying to follow and learn from the figures.

We modified the legends and we hope they are clearer.

Finally, all the minor comments suggested by the reviewer were taken into account in the new version.

[revised manuscript text omitted]

---

## Referee Report (RR1)

**Review of "Breakup of nocturnal low-level stratiform clouds during the southern West African Monsoon Season"**

By: Zouzoua et al., Atmospheric Chemistry and Physics

**General Comments**

This is my second review of this manuscript. The authors have done an excellent job of addressing the majority of my comments and recommendations from the 1st review. I believe the authors have done a very good job of thoroughly addressing the other reviewer's comments as well, which further strengthened this manuscript. This manuscript tells a much clearer story, the results naturally flow from one to the next, and conclusions made throughout the paper are supported by the results and figures shown and also appropriately stated. I am especially impressed by the quality of the figures, and the addition of the new Figure 13 is particularly excellent. As with my original review, the scientific quality of this manuscript merits publication in ACP.

My only major concern with this manuscript remains its readability. This is a very long and highly detailed manuscript, and compared to other manuscripts of similar length and detail, this again took me several hours to read and re-read due to several English and grammatical-related issues. In my opinion, this is not a reason to hold this manuscript back from full publication, but I would highly encourage the authors to review my recommendations and seek an independent re-read and English/grammar check prior to final publication. The results and science contained within this manuscript are more than worthy of publication in ACP, however, I fear the readability of this manuscript in its present form will discourage and frustrate readers from reading this in its entirety. The vast majority of these issues were contained in Sections 1-4. Sections 5 and 6 were mostly fine.

I noticed many unnecessary uses of the word "the", and noted many instances in my specific comment suggestions where these extraneous words can be eliminated. This is ultimately minor from a writing standpoint, but this remains *highly* distracting as a reader. Please read through the text carefully and take care to improve the readability of this manuscript. I have made several suggestions as specific comments.

**Specific Comments**

P2, L12: "... (LLSC) are Earth's most common cloud type (Wood 2012)." I would rephrase this to say "... (LLSC) are one of Earth's most common cloud types (Wood 2012)." because cirrus clouds arguably cover more of the surface at a given time.

P2, L19: Drop "Indeed" from the beginning of the sentence. Also drop "the" before "the numerical simulations".

P2, L20: Drop "the" in front of "the meteorological conditions".

P2, L22: Drop "the" in front of "the forecasts of West African".

L2, L24: Do you mean to say weather observations over West Africa are scarce? A "weather monitoring network" on its own cannot be "scarce", but rather "limited". Please clarify.

P2, L28: "cold air associated to West African monsoon" change this to "cold air associated with the West African monsoon"

P2, L30: "However, the LLSC after the sunrise received little attention." change to "However, LLSC evolution after sunrise has received little attention by previous literature, and further motivates our present study."

P3, L4-5: suggested re-write: "A joint measurement campaign took place using airborne and ground-based platforms (Flamant et al., 2017; Kalthoff et al. 2018)"

P3, L6: I think you mean "example of" and not "overview of".

P3, L8: change "in order to conduct detailed study of the LLSC" to "for highly-detailed study of the LLSC"

P4, L6-8: I am very confused what this sentence means. What are the exact "roles" of horizontal advection and vertical wind shear in what exactly?

P4, L8: "after the sunrise" change to "after sunrise". Also add commas around " sunrise, which leads to the transition towards shallow convective clouds, has not..."

P4, L9: change "...this transition by the mean of idealized..." to "...this transition by using idealized..."

P4, L13: change "Our study aims at analyzing..." to "Our study analyzes..."

P4, L14: you can probably drop everything after "...DACCIWA experiment," as this information is already implied.

P4, L15: rewrite: "This study should provide complimentary guidance..."

P4, L16: change "follow" to "follows"

General comment: I did my best to track and suggest changes for wording changes. There are still far too many stylistic choices through this point of the manuscript that make it a chore to read through. The content is good otherwise, but I would encourage the authors to more thoroughly scrutinize and improve upon their writing style for more effective communication of these important scientific results.

P4, L20: change "... just before the sunrise, at..." to "...just before sunrise at...". The comma is not needed here.

P5, L5: I think you mean to say "typically characteristic" and not "typical characteristic".

P5, L8: minor addition, you should say "LLSC deck" instead of just "LLSC".

P5, L10-11: "... and ends at the cloud layer breakup..." just say "... and ends upon LLSC breakup...". There also appears to be extra spaces in your W m$^{-2}$ units... check this before submitting your final proof.

P5, L12: I would reframe "... on the stratocumulus dynamic is presented by..." as "... on stratocumulus properties and dynamics..." to better encapsulate the breadth of both studies you cited here. Also,

"Such a cloud…" is strange wording to me; I would just start the next sentences as "LLSC are regulated through…"

P5, L18: "… than they receive from the drier air above" could be improved to say "… than they absorb downwelling longwave radiation from the overlying atmosphere." Also replace "It is modulated…" by "Longwave cooling is modulated…" unless "It" is referring to something else?

P5, L20: "the" is used four times in this sentence. Please cut down on unnecessary uses of this word.

P5, L24: Suggested rewrite: "Precipitation formation, large-scale subsidence and entrainment typically warm and dry out the LLSC layer…."

P5, L31: "The processes-analyzed studies, …" I have never heard of "processes-analyzed" studies… I believe you mean to say "These process-level studies…" which is more commonly used within the cloud modeling community. I would also add that that the aforementioned citations you listed included a lot of data analysis from field campaigns, so saying "essentially based on numerical simulations" undermines the larger breadth of results within those studies. Please modify this part of the text to properly acknowledge this or clarify which studies do not have a field campaign or observational data-based analysis component to it.

P5, L33: Begin with "Over land, the main driver…". I would also recommend adding a citation at the end of this sentence.

P6, L1: Say "The LES developed by" rather than "The LES made by". Also change "prove an insight" to "provides insight".

P6, L3: do you mean to say cloud top radiative cooling is the "sole source term to the LWP budget"? The present wording is strange. I also presume you mean "the primary factor" instead of "the factor".

Sentence beginning at the end of P6, L3: full rewrite suggestion: "The breakup of the LLSC deck ~5 hours after sunrise is primarily due to a co-occurring decrease of cloud-top cooling and increase of cloud-top entrainment." No need to mention the effect on the LWP budget here, as this is implied.

P6, L8: do you mean the LES "is *initialized* with atmospheric and surface conditions"?

P6, L17: drop the word "undisturbed"

P6, L19: Whereabout in the troposphere was this anticyclonic vortex? "low troposphere" could imply near the surface, 700 mb or somewhere in between. Be more specific.

P6, L21: hyphen consistency – here you say "low-troposphere" but a few lines earlier you state "low troposphere". Take care to ensure this here and throughout the manuscript.

P7, L1: Surround "upon which our investigation is based" with commas. Also note "on" should be replaced with "upon".

P7, L2: You can remove the sentence beginning with "The instrumentation and the data…" and weave those references elsehwere in this section where you actually discuss the instrumentation and dataset.

P7, L7 "i.e. less than 1 mm" put this in parentheses and remove the commas.

P7, L17-20: You can probably shorten this sentence for clarity. Also, is the ceilometer capable of measuring multiple cloud layers when the underlying layer contains high liquid water path?

P8, L1: There is no need to mention that the radiosondes are "reusable". Also, is there a reason these soundings only achieved a maximum height of 1500 meters above ground level?

P8, L14: "the water deposition" just say "water deposition". Also, I think you mean to say "neither of these" rather than "neither these".

P9, L2: "corresponds to the convective time scale". Time scales for convection, at this point in the text, are not previously defined nor may they be well known to the reader. I would state here or earlier in the text what time scales are typical for a full convection life cycle. Alternatively, you may want to state that the time averaging is done to better resolve processes throughout the process of convection.

P9, L25: You can eliminate "So that," and say "... may then be assessed..." later in this sentence.

P9, L27: change "by the use of" to "by using the"

P11, L10-11: Suggested rewrite: "The PREC term is typically near zero because no..."

P12, Equation 2: This equation seems out of place. Was this intended to be somwhere in the paragraph on P11, L8-14?

P12, L16: "lack of precise estimate" I would say "lack of a precise estimate"

P12, L25-26: I understand what you are trying to convey, but I would highly discourage saying there is "no possibility" of accurately estimating SUBS. I suggest rephrasing these sentence a bit, e.g. "...we cannot accurately estimate large scale subsidence...". Also I recommend rephrasing "consider evaluations" to "compute estimates".

P13, L11: Get rid of the word "Moreover," and change "consequently" to "thus".

P17, Figure 5: The grey edges are a bit difficult to see. I would make these thicker, or choose a different color that more easily contrasts with both the pink and black colors.

P18, L4: changes "decreases down to" to "decreases to".

P19, L8-9: Eliminate "In the next paragraph" and merge this sentence with the next paragraph.

P19, end of line 25: Will this be a topic of future study?

P20, Figure 7: Should the label in the figure read "08 July 06:21 UTC"?

P21, L1: Switch "In order to deepen the analysis," to "In this sub-section,"

P21, L14: change "... especially at the exit of the cloud." to "... especially near cloud top."

P21, L17: I recommended swapping "correct" with "accurate". Also, drop "measurements", this is implied in this sentence.

P22, L12: you can eliminate the words "in absolute."

P23, L4: What do you mean by "humidity jump"?

P23, L14-15: Is the "cloud layer" referencing the DACCIWA cases? Make this clear – the writing of this sentence implies the cloud layer refers to the van der Dussen case study. Also: say "on average" instead of "in average".

P23, sentence beginning on L15: You can eliminate this sentence. The discussion of the standard deviations here doesn't seem very important, especially considering it is "lower than 3% of the median".

P24, L1: I think you mean "falls" instead of "fits".

P24, L3: clarify "no particular difference" as "no significant difference".

P24, L15: change "this discussion shows" to "we show"

P25, L24: double check "$20.10^{-5}$ s^-2". Should the decimal point be a "x" sign? Also see L26.

P26, L15: change "by its own" to "on its own"

P26, L18: I think you mean to say "hypothesis cannot be fully tested".

P26, L23: I think you mean "warm air advection" rather than "warmer advection".

P26, L26: I would say "horizontal advection of colder air"

P26, L30-31: "maintains Rh constant" --> "maintains constant Rh". Also I could rephrase "the cases D" to "the decoupling cases". Finally, "temporal tendency" is redundant. Tendency implies time, so you can probably drop temporal, unless you mean to say "temporal variability" which could also make sense here.

P31, L5: just say "In the next sub-section,"

P34, L11: I would re-word the beginning of this sentence. Suggestion: "We define two periods, Ti and Tb, based on …"

P35, L2: I think you mean to say "objective of this study" and not "object of this study".

P36, L26: Given the area of focus for your study, you may be interested to read about the recent ORACLES field campaign that took place over the southeast Atlantic Ocean. The following reference offers an exhaustive description of the campaign with some preliminary analysis about aerosol types over this part of the world. The final two ORACLES deployments took place on the nearby island of Sao Tome.

Redemann, J., Wood, R., Zuidema, P., Doherty, S. J., Luna, B., LeBlanc, S. E., Diamond, M. S., Shinozuka, Y., Chang, I. Y., Ueyama, R., Pfister, L., Ryoo, J., Dobracki, A. N., da Silva, A. M., Longo, K. M., Kacenelenbogen, M. S., Flynn, C. J., Pistone, K., Knox, N. M., Piketh, S. J., Haywood, J. M., Formenti, P., Mallet, M., Stier, P., Ackerman, A. S., Bauer, S. E., Fridlind, A. M., Carmichael, G. R., Saide, P. E., Ferrada, G. A., Howell, S. G., Freitag, S., Cairns, B., Holben, B. N., Knobelspiesse, K. D., Tanelli, S., L'Ecuyer, T. S., Dzambo, A. M., Sy, O. O., McFarquhar, G. M., Poellot, M. R., Gupta, S., O'Brien, J. R., Nenes, A., Kacarab, M. E., Wong, J. P. S., Small-Griswold, J. D., Thornhill, K. L., Noone, D., Podolske, J. R., Schmidt, K. S., Pilewskie, P., Chen, H., Cochrane, S. P., Sedlacek, A. J., Lang, T. J., Stith, E., Segal-Rozenhaimer, M., Ferrare, R. A., Burton, S. P., Hostetler, C. A., Diner, D. J., Platnick, S. E., Myers, J. S., Meyer, K. G., Spangenberg, D. A., Maring, H., and Gao, L.: An overview of the ORACLES (ObseRvations of Aerosols

above CLouds and their intEractionS) project: aerosol-cloud-radiation interactions in the Southeast Atlantic basin, Atmos. Chem. Phys. Discuss., https://doi.org/10.5194/acp-2020-449, in review, 2020.

---

## Author Response (AR2)

Do you mean to say weather observations over West Africa are scarce? A "weather monitoring network" on its own cannot be "scarce", but rather "limited". Please clarify.

The sentence was modified, **_P2, L24-27_**:
*"Due to a limited weather monitoring network over West Africa, the first studies addressing LLSC over this region were mostly conducted with satellite images and traditional synoptic observations (Schrage and Fink, 2012; van der Linden et al., 2015), as well as with numerical simulations at regional scale (Schuster et al., 2013; Adler et al., 2017; Deetz et al., 2018)."*

I am very confused what this sentence means. What are the exact "roles" of horizontal advection and vertical wind shear in what exactly?

The sentence has been modified, and we hope it is now clearer (**_P4, L6-8_**):
*"They confirmed that the horizontal advection of colder air from the Guinean coast and mechanical turbulent mixing below the nocturnal low-level jet (NLLJ) are among the main drivers for LLSC formation."*

"The processes-analyzed studies, ..." I have never heard of "processes-analyzed" studies... I believe you mean to say "These process-level studies..." which is more commonly used within the cloud modeling community. I would also add that that the aforementioned citations you listed included a lot of data analysis from field campaigns, so saying "essentially based on numerical simulations" undermines the larger breadth of results within those studies. Please modify this part of the text to properly acknowledge this or clarify which studies do not have a field campaign or observational data-based analysis component to it.

The sentences have been modified as follow (**_P5, L30-32_**):
*"In these studies, the stratocumulus is initially coupled to the surface, with convective turbulence produced by the cloud-top radiative cooling. Specific mechanisms leading to the stratocumulus breakup are proposed, but are still based on an enhancement of the entrainment warming and drying effect."*

Do you mean to say cloud top radiative cooling is the "sole source term to the LWP budget"? The present wording is strange. I also presume you mean "the primary factor" instead of "the factor".

Sentence beginning at the end of P6, L3: full rewrite suggestion: "The breakup of the LLSC deck ~5 hours after sunrise is primarily due to a co-occurring decrease of cloud-top cooling and increase of cloud-top entrainment." No need to mention the effect on the LWP budget here, as this is implied.

The statement has been corrected as follows (*P6, L2-5*):
*"Before sunrise, the longwave radiative cooling at the LLSC top is the sole source term of the LWP budget and the primary factor maintaining this cloud layer. The breakup of the LLSC deck five hours after sunrise is primarily due to a decrease of cloud-top radiative cooling together with an increase of cloud-top entrainment."*

Drop the word "undisturbed"

We do think that this word is important. It specifies that the conditions in the monsoon layer before and after 08 July 2016 are not the same (*P6, L16*).

Whereabout in the troposphere was this anticyclonic vortex? "low troposphere" could imply near the surface, 700 mb or somewhere in between. Be more specific.

The sentence has been modified as follow (*P6, L17-19*):
*"Between 9 and 16 July 2016, the formation of nocturnal LLSC over SWA was inhibited by drier conditions in the monsoon layer due to an unusual anticyclonic vortex (identified at 850 hPa)."*

You can probably shorten this sentence for clarity. Also, is the ceilometer capable of measuring multiple cloud layers when the underlying layer contains high liquid water path?

This sentence is now revised and following the suggestions, additional information was added (*P7, L14-16*). Also, the measurement of higher cloud base height can be inaccurate when the underlying cloud layer contains high liquid water path. However, we use the first detected cloud base height by the ceilometer which is not impacted by signal attenuation.

There is no need to mention that the radiosondes are "reusable". Also, is there a reason these soundings only achieved a maximum height of 1500 meters above ground level?

We use the word "reusable" to be consistent with the previous DACCIWA research work based on the Savè supersite (Adler et al., 2019; Babić et al., 2019; Lohou et al., 2020), and to mark the difference between 'standard' radiosondes and 'reusable' radiosondes, which do not supply the same meteorological profile at the end (different altitude reached). The reason for which the reusable achieved only a maximum height of 1.5 km a.g.l was already indicated, but the statement has been modified to make it clearer (*P7, L28-31*):

*"In between these soundings, so-called "reusable" radiosondes were launched more frequently, at regular time intervals. At the height of 1.5 km a.g.l, the reusable radiosonde is released from its ascending balloon, falls at the surface within a reasonable distance to be easily found and used again (Legain et al., 2013). This system allowed providing a higher temporal resolution of the conditions within the monsoon layer."*

"corresponds to the convective time scale". Time scales for convection, at this point in the text, are not previously defined nor may they be well known to the reader. I would state here or earlier in the text what time scales are typical for a full convection life cycle. Alternatively, you may want to state that the time averaging is done to better resolve processes throughout the process of convection.

The sentence has been modified (***P9, L1-2***):
*"The diagnostics are calculated over a time interval of 10 minutes with a moving window of 5 minutes, which is suitable for resolving the processes-related to convection."*

Will this be a topic of future study?

The study of Pedruzo-Bagazgoitia et al. (2020) already demonstrated that the contribution of surface turbulent fluxes to LLSC dynamic is negligible during the night. The sentence has been corrected (***P19, L24-25***):
*"This may be related to the negligible contribution of surface fluxes during the stratus phase (Pedruzo-Bagazgoitia et al., 2020)."*

What do you mean by "humidity jump"?

The sentence is now (***P22, L17-P23, L1***):
*"The vertical profile used by Pedruzo-Bagazgoitia et al. (2020) to initialize their LES had a $\Delta\theta_l$ of 4.5 K and no jump of $q_t$ across the LLSC top."*

Is the "cloud layer" referencing the DACCIWA cases? Make this clear – the writing of this sentence implies the cloud layer refers to the van der Dussen case study. Also: say "on average" instead of "in average".

The sentence is now (***P23, L11-13***):

[revised manuscript text omitted]